# Century-scale wood nitrogen isotope trajectories from an oak savanna with variable fire frequencies

Matthew L. Trumper[1], Daniel Griffin[1], Sarah E. Hobbie[2], Ian M. Howard[3], David M. Nelson[4], Peter B. Reich[5,6], and Kendra K. McLauchlan[7]

[1]Department of Geography, Environment, and Society, University of Minnesota, Minneapolis, MN 55455, USA ORCiD identifiers 0000-0002-9881-7742 and 0000-0002-1547-3615
[2]Department of Ecology, Evolution, and Behavior, University of Minnesota, Saint Paul, MN 55108, USA ORCiD identifier 0000-0001-5159-031X
[3]Department of Geosciences, University of Arkansas, Fayetteville, AR 72701, USA ORCiD identifier 0000-0002-8361-9575
[4]Appalachian Laboratory, University of Maryland Center for Environmental Science, Frostburg, MD 21532, USA ORCiD identifier 0000-0003-2755-5535
[5]Department of Forest Resources, University of Minnesota, St Paul, MN 55108, USA ORCiD identifier 0000-0003-4424-662X
[6]Hawkesbury Institute for the Environment, Western Sydney University, Penrith, New South Wales 2753, Australia
[7]Department of Geography and Geospatial Sciences, Kansas State University, Manhattan, KS 66506, USA ORCiD identifier 0000-0002-6612-1097

**Correspondence:** Matthew L. Trumper (trump022@umn.edu)

**Abstract.** Fire frequency exerts a fundamental control on productivity and nutrient cycling in savanna ecosystems. Individual fires often increase short-term nitrogen (N) availability to plants, but repeated burning causes ecosystem N losses and can ultimately decrease soil organic matter and N availability. However, these effects remain poorly understood due to limited long-term biogeochemical data. Here, we evaluate how fire frequency and changing vegetation composition influenced wood stable N isotopes ($\delta^{15}$N) across space and time at one of the longest running prescribed burn experiments (established in 1964). We developed multiple $\delta^{15}$N records across a burn frequency gradient from precisely dated *Quercus macrocarpa* tree-rings in an oak savanna at Cedar Creek Ecosystem Science Reserve, Minnesota, USA. Sixteen trees were sampled across four treatment stands that varied in temporal onset of burning and burn frequency, but were consistent in overstory species representation, soil characteristics, and topography. Burn frequency ranged from an unburned control stand to a high fire-frequency stand that burned in four of every five years during the past 55 years. Because N stocks and net N mineralization rates are currently lowest in frequently burned stands, we hypothesized that wood $\delta^{15}$N trajectories would decline through time in all burned stands, but at a rate proportional to fire frequency. We found that wood $\delta^{15}$N records within each stand were remarkably coherent in their mean state and trend through time. A gradual decline in wood $\delta^{15}$N occurred in the mid 20th century in the no-, low-, and medium-fire stands, whereas there was no trend in the high-fire stand. The decline in the three stands did not systematically coincide with the onset of prescribed burning. Thus, we found limited evidence for variation in wood $\delta^{15}$N that could be attributed directly to long-term fire frequency in this prescribed burn experiment in temperate oak savanna. Our wood $\delta^{15}$N results may instead reflect decadal-scale changes in vegetation composition and abundance due to early to mid 20th century fire suppression.

# 1 Introduction

Fire is a fundamental control of species composition, diversity, and nutrient cycling in savanna ecosystems. Fire affects N pools and cycling in myriad ways that vary spatially and temporally (Pellegrini et al., 2015). On timescales of days to years, fire often enhances N availability to plants – possibly due to direct ash deposition or increases in microbial mineralization (Wilson et al., 2002; Boring et al., 2004). Furthermore, the loss of plant biomass during fire reduces plant N uptake, thereby increasing inorganic N pools (Ficken and Wright, 2017). In the long-term (decades to centuries), fire may reduce ecosystem N stocks due to elevated N volatilization and leaching losses, as well as diminished plant biomass (Raison, 1979; Ojima et al., 1994; Reich et al., 2001). Changes in vegetation composition may also influence the trajectory of soil N biogeochemistry, as the production of plant litter with low N concentration can slow decomposition and net N mineralization, reducing N supply and inorganic N pools (Ojima et al., 1994; Dijkstra et al., 2006). However, there remains uncertainty regarding multi-year to multi-decadal effects of fire on N cycling in savanna ecosystems because there is a paucity of long-term records of N cycling from areas with different fire-return intervals (Reich et al., 2001; Coetsee et al., 2008).

In the absence of long-term baseline biogeochemical data, the ratio of $^{15}$N to $^{14}$N (i.e., $\delta^{15}$N) in wood from trees can provide a proxy of N pools and cycling that extends decades or centuries into the past (Gerhart and McLauchlan, 2014). Plant $\delta^{15}$N can be a useful metric of plant N availability, defined as soil N supply relative to plant demand for N (Craine et al., 2015). When N availability is high, N is cycled and lost primarily as inorganic N, and relatively high rates of processes such as nitrification, nitrate leaching, ammonia volatilization, and denitrification lead to increased $\delta^{15}$N values of remaining N pools (Craine et al., 2015; van der Sleen et al., 2017). When N availability is low, reduced N losses result in lower $\delta^{15}$N values of remaining N pools. In addition, plants are more likely to receive N from mycorrhizal fungi than from direct uptake from inorganic N pools; mycorrhizal fungi are known to provide N with relatively low $\delta^{15}$N values to plants (Hobbie and Högberg, 2012). Prior studies have used natural-abundance measurements of wood $\delta^{15}$N to better understand the consequences of natural and human-induced disturbance to the N cycle, such as changes in N cycling due to fire (Beghin et al., 2011; Kranabetter and Meeds, 2017), tree-clearing (Bukata and Kyser, 2005; Hietz et al., 2010), and changing management practices (Howard and McLauchlan, 2015). These studies demonstrate the ability of tree-ring $\delta^{15}$N values to capture the role of disturbance and land-use change in altering N cycling at local scales.

Demonstrated effects of fire on N cycling (Pellegrini et al., 2015) and ability of wood $\delta^{15}$N to capture variation in N cycling suggest that long-term wood $\delta^{15}$N records can provide insight on past effects of fire on N cycling. Yet few studies have focused on wood $\delta^{15}$N response to fire, and thus the relative importance of fire-related processes that affect wood $\delta^{15}$N remain poorly understood. On the one hand, increased N volatilization and combustion of surface soils that typically have relatively low $\delta^{15}$N values are expected to increase plant $\delta^{15}$N values after fire events (Högberg, 1997). Alternatively, by reducing N stocks and rates of cycling, frequent fire might reduce non-fire N losses (e.g., via leaching, denitrification, and/or ammonia volatilization) and thereby decrease plant $\delta^{15}$N values. The net effect of these opposing effects of fire on $\delta^{15}$N values likely depends on the degree to which fire leads to isotopic fractionation associated with combustion processes versus N loss pathways (Fig. 1). For example, incomplete combustion of plant biomass can cause volatilization of NH$_3$, thereby increasing $\delta^{15}$N in the remaining

pool (Pellegrini et al., 2014). Strong isotopic fractionations associated with gaseous loss pathways can lead to N pools with relatively high $\delta^{15}N$ values (Houlton et al., 2006), and nitrification followed by leaching of nitrate can also cause higher $\delta^{15}N$ values of the remaining N (Gerhart and McLauchlan, 2014). Therefore, fire-induced "tightening" of the N cycle through reductions in non-fire N losses might only alter plant $\delta^{15}N$ values if gaseous losses are high in the unburned ecosystem. Beghin et al. (2011) observed increases in wood $\delta^{15}N$ in the five years after a large fire, consistent with isotopic fractionation effects during combustion, but 6–10 years after fire $\delta^{15}N$ values returned to pre-disturbance levels. Field studies show both increases and decreases in soil $\delta^{15}N$ in days to decades after fire, underlining the importance of post-disturbance processes that shape ecosystem $\delta^{15}N$ (Grogan et al., 2000; Aranibar et al., 2003; Perakis et al., 2015).

In this study, we use one of the longest running savanna burn experiments worldwide to evaluate how frequent fire influenced wood $\delta^{15}N$ as an indicator of effects on N availability across space and time. The experiment is located at Cedar Creek Ecosystem Science Reserve (CCESR), a Long Term Ecological Research (LTER) site in eastern Minnesota, USA where 55 years of prescribed fire have been linked to substantial declines in N stocks, net N mineralization rates, and litter N cycling in frequently burned stands (Reich et al., 2001). We examined wood $\delta^{15}N$ patterns across four oak woodland/forest and savanna plots that were similar in overstory species representation, soil characteristics, and topography in an ongoing long-term controlled burn experiment at CCESR. We hypothesized that the mean and trajectory of wood $\delta^{15}N$ values would be coherent (i.e., vary simultaneously) across treatment stands before the onset of prescribed fire, whereupon wood $\delta^{15}N$ values would decrease thereafter in burned plots at rates proportional to the frequency of prescribed fire.

## 2  Methods

### 2.1  Study Site

CCESR in eastern Minnesota is located within the continental-scale ecotone between the northern Great Plains prairie, the eastern deciduous woodlands, and the southwestern extent of the northern mixed conifer forests. Many upland and lowland plant communities exist at CCESR. Of particular interest is the oak savanna, which was once an important ecosystem at the prairie-forest border (Nuzzo, 1986), but now survives as a relatively rare component of the post European-settlement landscape. The oak savanna terrain has low relief with sandy, excessively drained soils that are N-poor (Udipsamments, Grigal et al., 1974; Tilman, 1984). Throughout our wood nitrogen isotope record spanning 1902-2017, mean annual temperature was 4.8ºC and mean annual precipitation was 72 cm. Wet plus dry atmospheric N deposition inputs at CCESR are ~ 4–10 kg N·ha$^{-1}$·yr$^{-1}$ (NADP, 2018). Annual $NH_4^+$ inputs have marginally increased and $NO_3^-$ inputs have marginally decreased between 1997-2017. Combined, total N deposition inputs at CCESR have largely remained stable since data collection began in 1984. Fire suppression was the management practice within the reserve from at least 1938–1964; however, this area burned periodically before and after Euro-American settlement in the late 19th century (Leys et al., 2019).

The burn experiment at the oak savannas of CCESR was established in 1964 to preserve vegetation communities and to investigate ecosystem response to fire treatment (Peterson and Reich, 2001; Reich et al., 2001). Nineteen units of 2.4 to 30 ha were each assigned a burn frequency treatment, ranging from fire exclusion to a four out of five year burn frequency (Fig. 2).

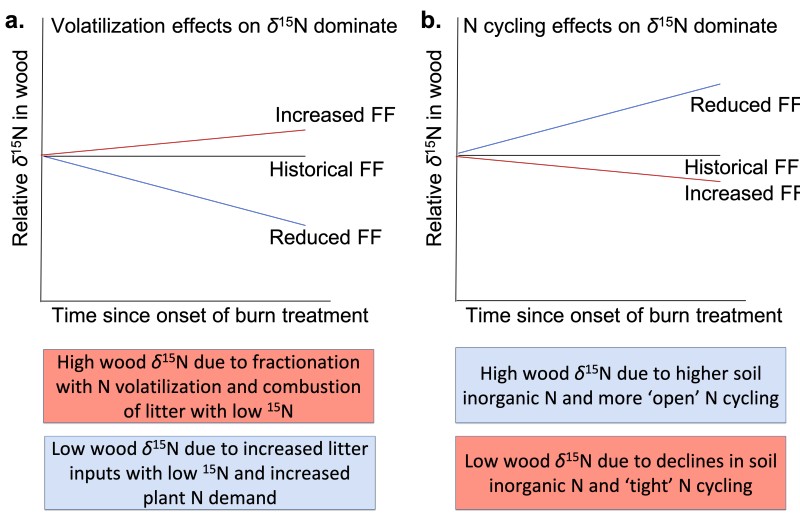

**Figure 1.** Hypothesized effects of fire experimentation on wood $\delta^{15}N$ values relative to the historical fire frequency (FF). Increased FF would lead to higher relative wood $\delta^{15}N$ values if combustion and isotopic fractionation with N volatilization dominate (a). In contrast, increased FF would lead to lower relative wood $\delta^{15}N$ values if N cycling effects such as reduced N stocks and non-fire N losses dominate the trajectory of wood $\delta^{15}N$ values (b). Colored boxes indicate potential contrasting effects of fire frequency on wood $\delta^{15}N$ values. Pink and blue boxes indicate processes resulting from higher and lower FF, respectively.

Prescribed burns typically occurred in April or May and fires were of low intensity. Although systematic vegetation inventories were not conducted prior to the onset of prescribed burning, a series of historical aerial photographs are available for our study area between the years 1938 and 2016 (McAuliffe et al., 2017; MHAPO, 2019). Our qualitative interpretation of these photographs indicates that the savanna vegetation prior to fire treatment at CCESR consisted of mixed woodland and grass communities (Fig. 2). Over time, oak tree vegetation – predominantly bur oak, *Quercus macrocarpa*, and northern pin oak, *Quercus ellipsoidalis* – increasingly dominated fire-excluded stands, while $C_4$ grasses and sedges increased in abundance in frequently burned stands (Peterson and Reich, 2001; Dijkstra et al., 2006). The cumulative effects of direct fire-induced N losses and indirect fire-induced shifts in vegetation composition since 1964 have caused the vegetation and soil characteristics of these burn stands to diverge dramatically (Fig. 3). Unburned stands are characterized by high oak dominance, increased woody plant biomass (including mesic taxa such as *Acer*), and high rates of soil N cycling and productivity, whereas frequently burned stands show high grass dominance, scattered bur and northern pin oak, and low rates of N cycling and productivity (Reich et al., 2001). We selected four stands for this study to span the gradient of fire frequency (from unburned to burned every 4 in 5 years) and the temporal onset of the fire experiment.

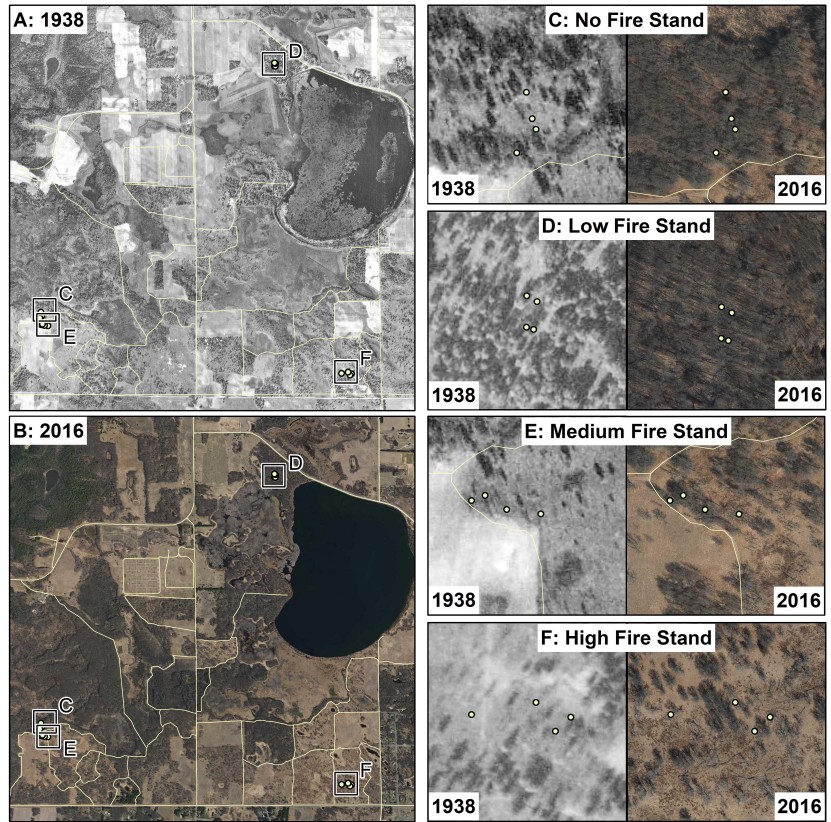

**Figure 2.** Historical aerial imagery for 1938 (A) and 2016 (B) for the savanna burn experiment and locations for 16 trees analyzed for wood $\delta^{15}$N at 1:50,000. Polygons illustrate roads at CCESR. Panels C - F illustrate stand level differences in canopy openness from 1938 to 2016 across the four fire treatments at a scale of 1:5,000. Imagery from Minnesota historical aerial photographs online (MHAPO, 2019).

## 2.2 Sample Collection

Increment cores were collected within ~1 m of the root collar from 16 *Q. macrocarpa* trees at CCESR in March 2018 with a 5.15 mm diameter Haglöf borer. *Q. macrocarpa* was the only tree sampled for wood N isotope analysis because this is the dominant overstory species (in age and frequency) of the savanna where the burn experiment is located. Sampling a single species also allowed us to control for potential species effects on wood $\delta^{15}$N. Trees spanned four stands in the fire frequency experiment: a no-fire stand (control), a low fire-intensity stand burned every 1 in 10 years (BU113), a medium fire-intensity stand burned every 1 in 3 years (BU115), and a high fire-intensity stand burned every 4 in 5 years (BU104). The onset of burning in the fire treatment stands was 1968, 1992, and 1965, respectively. Prior to that, fire episodes in a nearby stand had a median return interval of 15 years between 1822–1924, followed by a period of localized suppression (Leys et al., 2019). Broad soil characteristics were similar across sampled stands. Trees in the low-fire stand grew on Sartell fine sand mapped in soil surveys with 0–6% slopes and trees in no-, medium-, and high-fire stands grew on Sartell fine sand mapped in soil surveys with

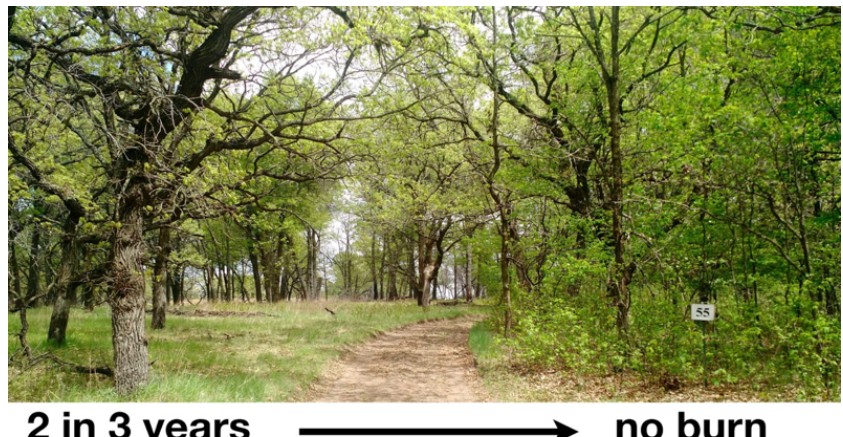

**Figure 3.** The Cedar Creek Ecosystem Science Reserve burn experiment. The difference in vegetation between unburned and frequently burn treatments stands is clearly shown. On left, a frequently burned stand is characterized by increased canopy openness and higher grass dominance. On right, an unburned control stand is characterized by dense undergrowth and high oak dominance. Photo credit: Peter Reich.

6–15% slopes. In the localized vicinity of the trees sampled across these sites, slopes rarely exceeded 8%. The Sartell series consists of excessively drained soils that are rapidly permeable and have low available N, low organic matter content, and low available water capacity (Grigal et al., 1974). The upper 15 cm of Sartell fine sand has a pH of 5.3, 0.025% total nitrogen (18 $\mu$mol/g dry soil), and ~0.3% organic matter (Grigal et al., 1974; Tilman, 1984).

Following sampling, the cores were stored in paper straws and dried at 65º C. After drying, each core was surfaced using a core microtome, visually cross-dated (Stokes and Smiley, 1996), and measured with micrometry. The pith was within 25 rings in 13 out of 16 samples. The mean series length was 157 years, with the oldest tree containing 252 rings and the youngest containing 109 rings (Table 1). Mean ring width from 1902–2017 across all N cores was 1.11 mm. Cores for $\delta^{15}$N analysis were cut into 10-mg samples using a razor blade along annual ring boundaries. Samples on average contained 1.35 years of

wood; 70% of samples contained only one year and 95% of samples contained one or two years. Although wood samples were partitioned at annual resolution, we did not analyze all wood samples for $\delta^{15}$N due to cost and time constraints. Rather, we selected two wood samples per decade for $\delta^{15}$N measurement. The decision to forego annual resolution and analyze our data at supra-annual timescales also aimed to mitigate the known phenomenon of inter-ring mobility of N-containing compounds that could smooth out inter-annual variation in $\delta^{15}$N (Hart and Classen, 2003; McLauchlan et al., 2017). N translocation has

long been recognized as an issue in interpreting wood $\delta^{15}$N. We did not perform chemical pretreatment on wood samples, as no pretreatment protocols have been widely shown to remove labile N in wood (Sheppard and Thompson, 2000; Bunn et al., 2017).

        Wood samples were analyzed for $\delta^{15}$N and wood % N at the Central Appalachians Stable Isotope Facility (CASIF) of the University of Maryland Center for Environmental Science. Samples were combusted in a Carlo Erba NC2500 elemental

analyzer (CE Instruments). Carbosorb and $MgClO_4$ were used to absorb $CO_2$ and $H_2O$, respectively, prior to the introduction

of $N_2$ to a Thermo Fisher Delta V+ isotope ratio mass spectrometer for $\delta^{15}N$ analysis. The $\delta^{15}N$ data were normalized to the AIR scale using a two-point normalization curve with internal standards calibrated against USGS40 and USGS41. Analytical precision for $\delta^{15}N$ of an internal wood standard analyzed along with samples was 0.3 ‰. A mass series of an atropine standard was used to calculate % N. In total, 367 wood samples were analyzed for $\delta^{15}N$ and % N across 16 trees. Samples correspond to the second and seventh calendar year in each decade and temporal coverage spans 1902-2017.

| Burn Unit | Burn Stand | Tree | Specimen ID | Ring Counts | Tree Age Class | First $\delta^{15}$ Sample Year | # $\delta^{15}$ N Samples | Fire Median Return (Years) | Burning Onset (Year) |
|---|---|---|---|---|---|---|---|---|---|
| Control | No-fire | 1 | CCS033 | 109 | 1 | 1927 | 19 | 0 | NA |
| Control | No-fire | 2 | CCS034 | 132 | 2 | 1937 | 17 | 0 | NA |
| Control | No-fire | 3 | CCS038 | 107 | 1 | 1917 | 21 | 0 | NA |
| Control | No-fire | 4 | CCS071 | 131 | 2 | 1927 | 19 | 0 | NA |
| BU113 | Low-fire | 1 | CCS135 | 190 | 4 | 1902 | 24 | 10 | 1968 |
| BU113 | Low-fire | 2 | CCS136 | 197 | 4 | 1902 | 24 | 10 | 1968 |
| BU113 | Low-fire | 3 | CCS139 | 167 | 3 | 1902 | 24 | 10 | 1968 |
| BU113 | Low-fire | 4 | CCS141 | 206 | 4 | 1902 | 24 | 10 | 1968 |
| BU115 | Medium-fire | 1 | CCS236 | 129 | 1 | 1902 | 23 | 3 | 1992 |
| BU115 | Medium-fire | 2 | CCS271 | 150 | 2 | 1902 | 24 | 3 | 1992 |
| BU115 | Medium-fire | 3 | CCS273 | 152 | 2 | 1902 | 24 | 3 | 1992 |
| BU115 | Medium-fire | 4 | CCS274 | 161 | 3 | 1902 | 24 | 3 | 1992 |
| BU104 | High-fire | 1 | CCS545 | 116 | 1 | 1922 | 20 | 1 | 1965 |
| BU104 | High-fire | 2 | CCS556 | 144 | 2 | 1902 | 23 | 1 | 1965 |
| BU104 | High-fire | 3 | CCS558 | 161 | 3 | 1967 | 11 | 1 | 1965 |
| BU104 | High-fire | 4 | CCS559 | 252 | 4 | 1902 | 24 | 1 | 1965 |

**Table 1.** List of the trees sampled, ring counts, first sample year for $\delta^{15}N$, number of $\delta^{15}N$ samples, fire median return, and the first year of prescribed burning. Ring count is the total number of years for each core and does not represent specific tree age, as pith was not present for all cores. Trees in age class 1 contain <129 rings, trees in age class 2 contain 130-159 rings, trees in age class 3 contain 160-189 rings, and trees in age class 4 contain >190 rings.

## 2.3 Statistical Analysis

Beyond understanding essential differences in mean $\delta^{15}N$ values across burn stands, a primary goal was to understand whether varying fire treatment frequency and temporal onset of fire affected the mean and long-term trajectory of $\delta^{15}N$ values. We used several statistical methods to assess time series patterns of wood $\delta^{15}N$ and N cycling across the CCESR burn experiment. Although many studies standardize wood $\delta^{15}N$ values for each core with respect to that core mean (McLauchlan et al., 2007, 2017), we chose not to do so in order to maintain patterns that might otherwise be obscured with detrending or standardization

techniques. Differences in wood $\delta^{15}$N across burn treatment stands and individual trees were tested using the Kruskal-Wallis test, a nonparametric test to identify significant differences on a continuous dependent variable by a categorical independent variable. To test for evidence of long-term trends in $\delta^{15}$N, we used simple linear regression and the nonparametric Mann-Kendall trend test (MK test). The MK test looks for evidence of monotonic trend in a time series and is robust to non-normally distributed data. For linear regression and the MK test, trends were deemed significant for $p$ values $< 0.05$. In addition, we conducted regime shift detection analysis on $\delta^{15}$N values by burn stand to statistically identify changes in the mean state of $\delta^{15}$N following the method of Rodionov (2004). This method applies sequential $t$-tests to time series data to statistically identify shifts in the mean state (Rodionov, 2004). Significant shifts in mean $\delta^{15}$N were identified using the two-tailed Student $t$-test and the $\alpha = 0.01$ threshold was used. A 6-year (1/4 series) cut-off length was used to allow flexibility in detecting mean shifts. The Huber's weight parameter was set to 1 and red noise was not estimated.

| Burn Unit | Burn Stand | Specimen ID | Tree Age Class | $m$ | $p$ value | Kendall's Tau |
|---|---|---|---|---|---|---|
| Control | No-fire | CCS033 | 1 | -0.19 | < 0.0001 | -0.70*** |
| Control | No-fire | CCS034 | 2 | -0.05 | NS | -0.23 |
| Control | No-fire | CCS038 | 1 | -0.25 | < 0.0001 | -0.85*** |
| Control | No-fire | CCS071 | 2 | -0.16 | < 0.0001 | -0.49** |
| BU113 | Low-fire | CCS135 | 4 | -0.20 | < 0.0001 | -0.74*** |
| BU113 | Low-fire | CCS136 | 4 | -0.21 | < 0.0001 | -0.54** |
| BU113 | Low-fire | CCS139 | 3 | -0.25 | < 0.0001 | -0.75*** |
| BU113 | Low-fire | CCS141 | 4 | -0.25 | < 0.0001 | -0.72*** |
| BU115 | Medium-fire | CCS236 | 1 | -0.20 | < 0.0001 | -0.82*** |
| BU115 | Medium-fire | CCS271 | 2 | -0.26 | < 0.0001 | -0.59*** |
| BU115 | Medium-fire | CCS273 | 2 | -0.22 | < 0.0001 | -0.78*** |
| BU115 | Medium-fire | CCS274 | 3 | -0.28 | < 0.0001 | -0.66*** |
| BU104 | High-fire | CCS545 | 1 | 0.02 | NS | 0.12NS |
| BU104 | High-fire | CCS556 | 2 | -0.03 | NS | -0.27NS |
| BU104 | High-fire | CCS558 | 3 | -0.02 | NS | -0.07NS |
| BU104 | High-fire | CCS559 | 4 | -0.03 | < 0.01 | -0.35* |

**Table 2.** Estimated slope coefficient based on linear regression ($m$), $p$-value based on linear regression, and Kendall's Tau based on the Mann Kendall test (MK test). The MK test looks for evidence of monotonic trend in a time series and is robust to non-normally distributed data. Tau values $*$ indicate $p < 0.05$, $**$ $p < 0.01$, and $***$ $p < 0.0001$; NS is not significant. Trees in age class 1 contain <129 rings, trees in age class 2 contain 130-159 rings, trees in age class 3 contain 160-189 rings, and trees in age class 4 contain >190 rings.

| Year | No-fire mean $\delta^{15}$N | Low-fire mean $\delta^{15}$N | Medium-fire mean $\delta^{15}$N | High-fire mean $\delta^{15}$N |
|---|---|---|---|---|
| 1902 | -2.19* | 0.38 | 1.65 | -1.16 |
| 1907 | -2.91* | 0.54 | 2.21 | -1.58 |
| 1912 | -1.14* | 0.77 | 1.93 | -0.98 |
| 1917 | -0.03 | 1.19 | 1.94 | -0.46 |
| 1922 | -1.81 | 0.86 | 1.32 | -1.24 |
| 1927 | -0.7 | 0.98 | 1.37 | -0.09 |
| 1932 | -1.24 | 1.17 | 1.37 | -0.53 |
| 1937 | -0.65 | 0.94 | 1.25 | -0.66 |
| 1942 | -1.06 | 0.07 | 0.9 | -0.62 |
| 1947 | -1.27 | 1.26 | 0.26 | -0.5 |
| 1952 | -1.54 | -0.18 | -0.68 | -0.72 |
| 1957 | -1.63 | 0.05 | -0.35 | -0.09 |
| 1962 | -0.9 | -0.4 | -0.94 | -0.71 |
| 1967 | -2.34 | -2.2 | -2.07 | -0.6 |
| 1972 | -3.12 | -2.35 | -2.06 | -0.47 |
| 1977 | -3.15 | -2.54 | -1.94 | -0.67 |
| 1982 | -2.81 | -2.09 | -2.14 | -0.91 |
| 1987 | -2.6 | -2.45 | -2.25 | -1.24 |
| 1992 | -3.08 | -3.38 | -1.9 | -1.32 |
| 1997 | -3.28 | -2.65 | -2.29 | -1.33 |
| 2002 | -3.3 | -3.25 | -2.44 | -1.15 |
| 2007 | -3.2 | -3.04 | -2.41 | -0.93 |
| 2012 | -3.27 | -3.16 | -2.54 | -0.83 |
| 2017 | -3.53 | -2.87 | -2.16 | -1.03 |

**Table 3.** Mean wood $\delta^{15}$N (‰) by year in each burn unit. Mean values * indicate years with only one $\delta^{15}$N sample in that year and burn unit.

## 3   Results

To test whether wood $\delta^{15}$N values were broadly similar before fire and diverged after fire, we first examined spatial differences in $\delta^{15}$N. In the first decade of the 20th century, wood $\delta^{15}$N values varied among stands ($p = 0.023$). Average wood $\delta^{15}$N values in 1902 ranged from -2.2 to 1.7‰ among stands (Table 3). A Kruskal-Wallis test of wood $\delta^{15}$N across the time series for the four stands revealed that $\delta^{15}$N was significantly different among stands ($p < 0.001$). We used the nonparametric Games-Howell test for post-hoc analysis due to unequal variances and group sizes between burn treatment stands. The Games-Howell test revealed that $\delta^{15}$N in the no-fire stand was lower than all other stands ($p < 0.001$). In contrast, the Games-Howell test revealed nonsignificant differences in $\delta^{15}$N between the low- and medium-fire stands ($p = 0.385$), low- and high-stands ($p = 0.944$), and

medium- and high-fire stands ($p = 0.441$). Wood $\delta^{15}$N was also significantly different between stands prior to the onset of the prescribed burn experiment in 1964 ($p < 0.001$). Wood $\delta^{15}$N differences among individual trees were small within each stand; only tree CCS545 in the high-fire stand was significantly different from the other three trees in that stand ($p < 0.001$). Two approaches revealed that tree age likely did not have an effect on the mean or trend of $\delta^{15}$N. First, we compared wood $\delta^{15}$N

with tree biological age rather than Gregorian calendar year using inner ring dates to estimate tree age. Tree age and $\delta^{15}$N were negatively correlated ($r = -0.35$; $p < 0.001$), however this negative relationship may be expected given the negative temporal trends in $\delta^{15}$N in the majority of trees. Additionally, we tested for an age effect on wood $\delta^{15}$N by dividing trees into four age classes based on ring counts (Table 1, Table 2). A Kruskal-Wallis test of wood $\delta^{15}$N across age classes revealed that tree age did not have a significant effect on the mean ranks of $\delta^{15}$N ($p = 0.242$).

Over time and in aggregate, wood $\delta^{15}$N values have declined at CCESR since approximately 1967, although specific trajectories varied by fire frequency (Fig. 4, Table 2). Negative trends in wood $\delta^{15}$N were evident in 12 out of 16 trees sampled, 11 of which were in the no-fire, low-fire, and medium-fire stands. Declines in wood $\delta^{15}$N occurred in the mid 20th century in these stands, beginning primarily between 1940 and 1965. In contrast, there was little evidence of declining wood $\delta^{15}$N in the high-fire stand, where $\delta^{15}$N values were relatively stable over time. Results from the regime-shift analysis were consistent with

the trend tests; shifts in the mean state of $\delta^{15}$N were detected in the no-fire, low-fire, and medium-fire frequency stands ($p < 0.01$), whereas there was not a $\delta^{15}$N shift in the high-fire stand at the $\alpha = 0.01$ significance level (Fig. 5).

     In the no-fire stand, three out of four sampled trees had significantly declining trends in wood $\delta^{15}$N to present based on simple linear regression ($p < 0.0001$) and the MK test ($p < 0.01$; Fig. 4a). Although the wood $\delta^{15}$N trend for tree CCS034 was not significant, this tree had a negative $\delta^{15}$N slope that matched the overall negative trend in this stand. Regime shift analysis

detected one mean shift in the 1960s, reflecting a stable mean of higher wood $\delta^{15}$N between 1902 to 1962 followed by a stable mean of lower wood $\delta^{15}$N from 1967 to 2017. Wood $\delta^{15}$N in this stand had a maximum of 1.8 ‰, a minimum of -4.7 ‰, and standard deviation 1.3 ‰.

     In the low-fire stand, all four trees demonstrated significant declines in wood $\delta^{15}$N towards present based on the two trend tests ($p < 0.0001$; Fig. 4b). Regime shift analysis identified one mean wood $\delta^{15}$N shift in the 1960s, separating a stable mean

of higher wood $\delta^{15}$N between 1902 and 1962 and lower mean $\delta^{15}$N from 1967 to 2017. Fire treatment in this stand began in 1968 and with a 1 in 10-year fire median return, and this stand has burned five times throughout the experiment. The wood $\delta^{15}$N values ranged from 2.8 to -4.1 ‰ with standard deviation 1.9 ‰. There was tight coherence between individual cores during the period 1942–1962, including a steep decline in all four cores between samples 1962 and 1967. While this steep single increment decline bracketed the beginning of the burn experiment in 1964, prescribed fire had not yet occurred in this

stand.

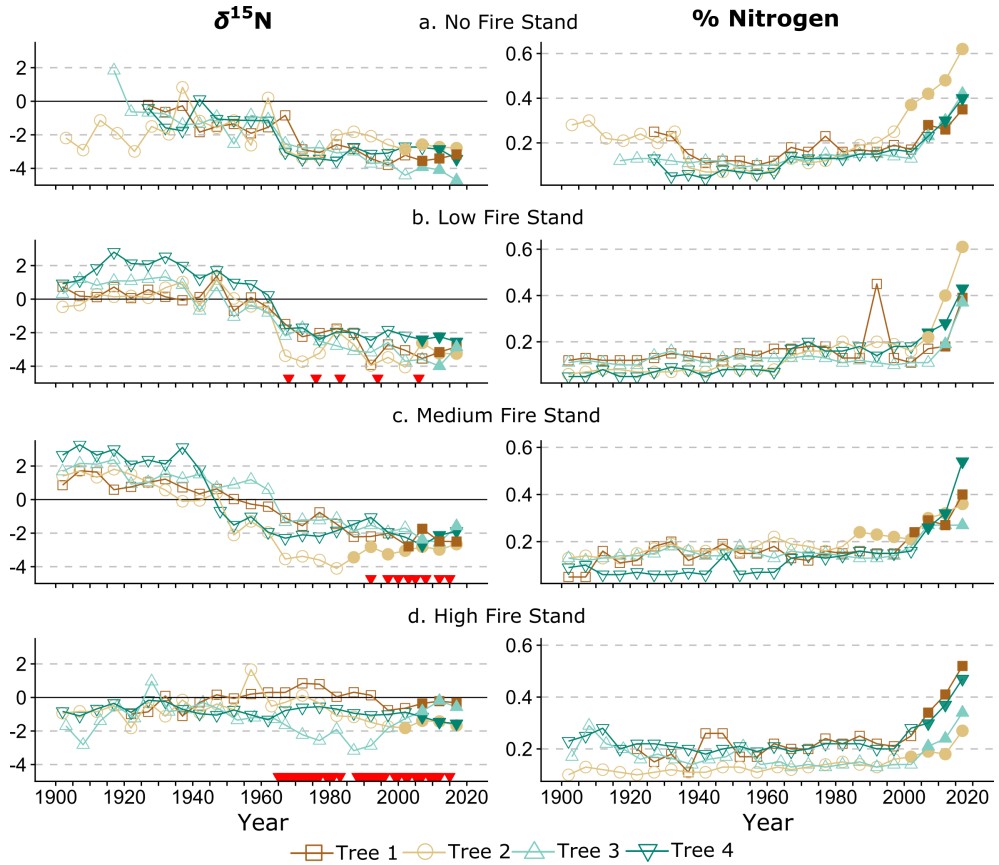

**Figure 4.** Wood $\delta^{15}$N (left) and % N (right) data over time. Lines connect data for individual trees within each stand. Shapes and colors correspond to trees 1–4 in each stand (Table 1). On left, red triangles indicate individual years of prescribed fire in each treatment stand. Filled shapes indicate wood sampled after the heartwood-sapwood transition. Wood % N increased in all trees over the last 115 years, with the sharpest increases after the heartwood-sapwood transition (average transition year 2004).

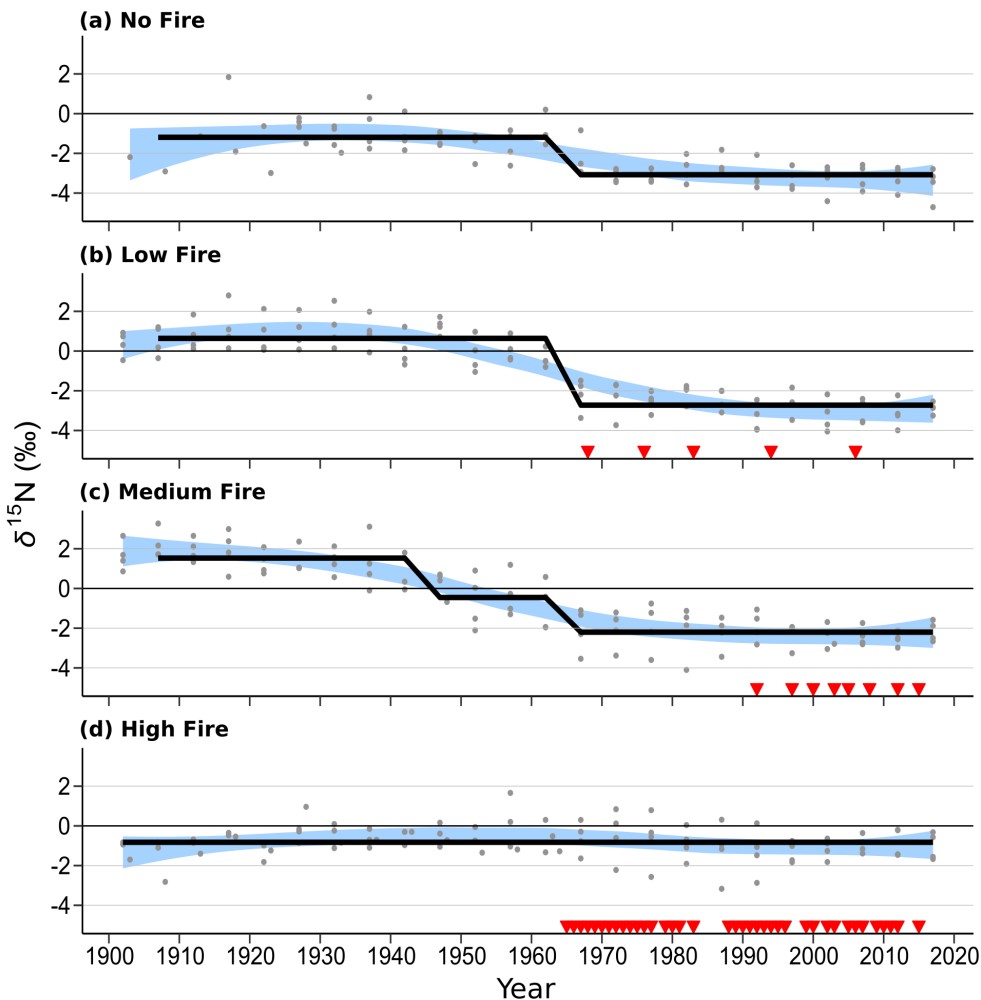

**Figure 5.** Wood $\delta^{15}$N patterns of mature *Quercus* trees in each of four burn stands. Four trees were sampled in each stand. Blue shaded areas represent 99% confidence intervals of a LOWESS curve fit and the thick black line shows mean $\delta^{15}$N states determined using regime shift analysis. Red triangles indicate individual years of prescribed fire in each treatment stand. Mid-20th-century declines in wood $\delta^{15}$N are readily seen in the no-fire, low-fire, and medium-fire stands.

In the medium-fire frequency stand, all four trees showed significantly declining trends toward present based on the two trend tests ($p < 0.0001$; Fig. 4c). Two mean shifts were detected in the medium-fire stand in the 1940s and 1960s using regime shift analysis. Fire treatment in this stand began in 1992 with a 1 in 3-year fire median return, totaling eight prescribed burns to the date of sampling. Trees in this stand exhibited the steepest declines in wood $\delta^{15}$N over time, ranging from a maximum of 3.3 ‰ in 1907 to a minimum of -4.1 ‰ in 1982. The standard deviation of wood $\delta^{15}$N values (1.9 ‰) was highest in this burn stand relative to all others.

In contrast to the other stands we sampled, the high fire frequency stand displayed little evidence for declining wood $\delta^{15}$N. Out of the four sampled trees, only CCS559 demonstrated a significantly declining trend in wood $\delta^{15}$N toward present based on the linear trend test (Fig. 4d). Similarly, no shifts in mean wood $\delta^{15}$N were detected using regime shift analysis. Fire treatment began in this stand in 1965 and has continued at a four in five-year annual fire return. In total, 38 fires occurred in this stand between 1965 and 2017, likely surpassing the pre-settlement fire frequency in this region (Leys et al., 2019). Wood $\delta^{15}$N values ranged from 1.7 to -3.2 ‰ and the standard deviation of $\delta^{15}$N (0.8 ‰) was lowest in this burn stand.

Wood % N increased in all trees over the last 115 years, with significant increases in 11 out of 16 trees based on the MK test ($p < 0.01$). A sharp increase in % N occurred after the heartwood-sapwood transition in most trees (Fig. 4). For these trees, the sapwood included the outer 10–20 years of growth, and the heartwood-sapwood transition consistently occurred in one ring and never in more than three. For each tree, % N increased consistently through the last sample, and the last two to three samples exhibited markedly higher % N than in previous years. This result is consistent with the well-documented pattern of increasing wood % N after the heartwood-sapwood transition with highest values in most recent rings (Poulson et al., 1995; McLauchlan and Craine, 2012). Wood % N was negatively correlated with wood $\delta^{15}$N across all trees ($r$ = -0.38; $p < 0.001$). Excluding all samples from the sapwood, wood % N and wood $\delta^{15}$N remained negatively correlated, statistically significant although the relationship weakened ($r$ = -0.30; $p < 0.001$). Small differences in wood % N trajectories among treatment stands were evident in our data and these differences were not statistically significant.

## 4   Discussion

*Q. macrocarpa* wood $\delta^{15}$N values were significantly different among sampled stands at the beginning of our 115-year record and before the onset of prescribed burning in 1964. Spatial differences indicate lack of support for the first part of our hypothesis, specifically that mean $\delta^{15}$N values would be similar across stands before the onset of the burn experiment. These differences in wood $\delta^{15}$N among stands suggest spatial heterogeneity in N availability before the onset of fire treatment. Over the full record, among-stand wood $\delta^{15}$N variability was higher than within-stand variability, potentially reflecting larger differences in vegetation composition and measured N cycling metrics between stands.

In contrast to wood $\delta^{15}$N, wood % N trajectories were positive and consistent among stands. Wood % N is not as commonly reported in wood N isotope studies because this metric likely incorporates tree physiology and thus may not reflect ecosystem N cycling (Gerhart and McLauchlan, 2014). Positive wood % N trajectories shown here match documented patterns of increasing wood % N after the heartwood-sapwood transition in the wood $\delta^{15}$N literature (Gerhart and McLauchlan, 2014), yet this pattern has seen more mixed results in the wood chemistry literature (Meerts, 2002; Martin et al., 2014). Although wood % N may not reflect N availability, coherence in wood % N patterns among individual trees and stands suggests that these data contain a strong signal that is independent from wood $\delta^{15}$N at this site. If this result reflects radial nitrogen translocation across ring boundaries, then the greatest mobility would have occurred within the last five years of growth, with a decay in mobility that extends over fifteen years. Annual-scale analyses might further elucidate these patterns. Wood % N and $\delta^{15}$N were negatively correlated across all samples and this negative relationship diminished when excluding samples from the sapwood. Although

we do not have further evidence to suggest that this is a meaningful relationship, future studies should examine relationships between wood % N and $\delta^{15}$N because their covariance could be problematic if trends in $\delta^{15}$N are strongly related to % N in addition to external factors.

Our results do not support the second part of our original hypothesis; wood $\delta^{15}$N values did not vary among stands proportional to fire frequency and known dates of fire and differences in fire frequency did not correspond to temporally consistent shifts in mean wood $\delta^{15}$N. The negative trend in wood $\delta^{15}$N for the majority of trees we analyzed matched our a priori expectation of declining wood $\delta^{15}$N based on evidence for long-term declines in rates of net N mineralization and tree productivity in frequently burned stands at CCESR (Reich et al., 2001; Dijkstra et al., 2006). However, wood $\delta^{15}$N was not inversely related to fire frequency. Trees in the low-fire frequency stands displayed significant negative trends in wood $\delta^{15}$N as expected. But trees in the no-fire stand showed similar declines in $\delta^{15}$N to that of trees in the low- and medium-fire stands, despite fire exclusion and higher measured N stocks and net N mineralization rates in this unburned stand toward present (Reich et al., 2001). Further, wood $\delta^{15}$N declines in the low- and medium-fire stand predated the onset of experimental prescribed burning, and trees in the high-fire stand demonstrated no significant positive or negative trend in $\delta^{15}$N in response to frequent fire beginning in 1965. Although regime shift analysis identified mean shifts that nearly aligned in time in the no-, low-, and medium-fire stands, all shifts preceded the onset of prescribed fire and overall $\delta^{15}$N declines were gradual. In combination with wood $\delta^{15}$N values in the high-fire stand showing no discernible response to fire, this evidence indicates that the onset of prescribed fire in the mid 20th century did not cause declines in wood $\delta^{15}$N at CCESR.

| Year | Data | Soil Depth (cm) | Low-fire | Medium-fire | High-fire |
|---|---|---|---|---|---|
| 1992 | Wood $\delta^{15}$N | NA | -2.46 | -1.06 | 0.13 |
| 1995 | Net N mineralization | 0–15 | 7.21 | 10.42 | 2.31 |
| 1997 | Wood $\delta^{15}$N | NA | -1.84 | -1.94 | -0.76 |
| 2016 | %N | 0–5 | 0.12 | NA | 0.1 |
| 2016 | %N | 5–10 | 0.06 | NA | 0.06 |
| 2016 | %N | 10–20 | 0.05 | NA | 0.06 |
| 2016 | %N | 20–60 | 0.02 | NA | 0.04 |
| 2016 | %N | 60–100 | 0.01 | NA | 0.03 |
| 2016 | $\delta^{15}$N | 0–5 | 1.13 | NA | 2.25 |
| 2016 | $\delta^{15}$N | 5–10 | 4.15 | NA | 3.57 |
| 2016 | $\delta^{15}$N | 10–20 | 4.76 | NA | 3.85 |
| 2016 | $\delta^{15}$N | 20–60 | NA | NA | 5.14 |
| 2016 | $\delta^{15}$N | 60–100 | NA | NA | 2.93 |
| 2017 | Wood $\delta^{15}$N | NA | -2.54 | -1.58 | -0.32 |

**Table 4.** Comparison of previously collected soil N data and our wood $\delta^{15}$N data from CCESR. Values represent the mean of all samples within each burn unit. Net N mineralization data from Reich et al. (2001), and soil % N and $\delta^{15}$N data from Pellegrini et al. (2020).

Prior studies at CCESR have examined N cycling in the same stands we sampled for wood $\delta^{15}$N across the burn experiment. Reich et al. (2001) measured net N mineralization in 1995 in three of the four stands we sampled (low-, medium-, and high-fire). Net N mineralization decreased nonlinearly with increasing fire frequency among these stands. However, our wood $\delta^{15}$N samples in 1992 and 1997 increased with increasing fire frequency among these stands. Wood $\delta^{15}$N values from 1992 and 1997

were lower in the low- and medium-fire stands compared to the high-fire stand, despite relatively higher net N mineralization in the former stands in 1995 (Table 4; Reich et al., 2001). This could point to large inter-annual variation in wood $\delta^{15}$N or that soil net N mineralization may not be strongly related to wood $\delta^{15}$N at our study site. More recently, Pellegrini et al. (2020a) measured soil $\delta^{15}$N and cumulative N stocks at multiple depths in two of the four stands we sampled (low- and high-fire) in 2016. They found $\delta^{15}$N decreased with more frequent burning and $\delta^{15}$N increased with depth across the entire fire frequency

gradient, while fire did not have a clear effect on total soil N between the low- and high-fire frequency stands we sampled (Pellegrini et al., 2020b). In contrast, our wood $\delta^{15}$N in 2017 was significantly lower in the low-fire stand compared to the high-fire stand and we found limited evidence for variation in wood $\delta^{15}$N related to long-term fire frequency. Altogether, these nuanced results from previous studies add context but do not clarify interpretation of our observed wood $\delta^{15}$N patterns.

Based on the lack of evidence for direct fire effects on short and long-term wood $\delta^{15}$N patterns at this site, we believe the

primary driver of $\delta^{15}$N patterns at this site has been changes in vegetation composition and abundance over time, perhaps driven in part by early to mid 20th century fire suppression. Our qualitative assessment of aerial photographs between 1938 and 2016 indicates that canopy density increased through time in the no-, low-, and medium-fire stands but remained relatively constant in the high-fire stand (Fig. 2). Our interpretation is in general agreement with results from other studies at CCESR (Peterson and Reich, 2001), and a directly adjacent experimental site (Faber-Langendoen and Davis, 1995) with the same

soil, topography, and plant community characteristics. If fire suppression during several decades of this period indeed allowed understory shrubs and young trees to increase in abundance in these plots, total plant N demand likely increased, thereby reducing N availability to mature oaks. In addition, increased litter inputs with low $^{15}$N would tend to decrease soil $\delta^{15}$N in these stands (Nadelhoffer and Fry, 1988). These processes would explain mid-century wood $\delta^{15}$N declines in these plots, particularly in the low- and medium-fire stands where $\delta^{15}$N declined most sharply. In contrast, the high-fire plot was more

grass-dominated and open relative to surrounding stands in 1938 and has remained so over time. Less ingrowth of understory shrubs and young trees in the high-fire stand could result in low total N demand that was stable over time, perhaps accounting for the lack of negative $\delta^{15}$N trend in this stand. Fire suppression was also suggested as a driver of declining N availability based on wood $\delta^{15}$N records in an old-growth pine forest in Minnesota (Howard and McLauchlan, 2015). Thus, although we did not find evidence for variation in wood $\delta^{15}$N directly related to prescribed fire, the indirect effects of fire suppression on

vegetation composition and abundance may have been sufficient to affect nitrogen resource partitioning between mature oaks and competing understory species, ultimately causing declining N availability to mature oaks in three out of four stands.

There are several alternative hypotheses to explain the lack of a direct relationship between prescribed fire and wood $\delta^{15}$N trajectories at CCESR. One hypothesis is that the declines in wood $\delta^{15}$N values in trees at CCESR may reflect a common continental-scale driver such as increased atmospheric $CO_2$ or N deposition (McLauchlan et al., 2017). However, such drivers

changed the most in the past 50 years, not in the 20 years prior when the largest changes in wood $\delta^{15}$N values occurred.

Moreover, the large differences in wood $\delta^{15}$N mean and trend between burn stands throughout our 115 record also suggest a minor role for continental-scale drivers at the scale of our study. The relative close proximity of sampled trees within treatment stands may also contribute to the ambiguous relationship between fire and wood $\delta^{15}$N at this site. The four trees sampled per burn stand were spatially quite close to each other (on average 34 meters between trees within each burn unit), and we were unable to replicate sampling in other units with the same fire treatment. Given the spatial variability in wood $\delta^{15}$N that existed before the fire treatment began, additional replicates of fire treatment would help to clarify the relationship between fire and wood $\delta^{15}$N.

Our results are more likely the result of local-scale influences – apart from the obvious presence of fire – on N cycling and tree N uptake at CCESR. Despite our limited evidence for variation in $\delta^{15}$N of tree-rings due to prescribed fire, it is important to recognize that fire can influence $\delta^{15}$N and N availability through a variety of mechanisms. For example, Högberg (1997) proposed that the combustion of the upper $\delta^{15}$N-depleted surface layer would lead to increased $\delta^{15}$N values in plants shortly after fire. N losses from volatilization of N during fire events could exceed leaching losses here, given that bursts of plant N-uptake and growth after fire have been shown to limit leaching losses (Boerner, 1982; Dijkstra et al., 2006). However, evidence for this N volatilization mechanism may be limited at CCESR because fires in the burn experiment were low severity and rarely resulted in complete combustion of the litter layer (Hernández and Hobbie, 2008). Post-fire nitrification is another mechanism that could affect soil and wood $\delta^{15}$N, causing increased $\delta^{15}$N values of $NH_4^+$ and decreased $\delta^{15}$N values of $NO_3^-$ (Högberg, 1997). This mechanism could affect wood $\delta^{15}$N at this site, as there is evidence that *Quercus* seedlings show a greater relative preference for $NO_3^-$ over $NH_4^+$ (Templer and Dawson, 2004), and preferences for inorganic N were reflected in *Q. alba* wood $\delta^{15}$N in an Indiana hardwood forest (McLauchlan and Craine, 2012). Soil net nitrification was negatively related to fire frequency across the CCESR burn experiment (Reich et al., 2001) but our wood $\delta^{15}$N data cannot address the extent to which nitrification is altering the supply of available N. Beghin et al. (2011) invoked both litter combustion and post-fire nitrification in their study of long-term N cycling in a pine forest. They found increased wood $\delta^{15}$N in the 5 years after fire disturbance, but a return to pre-disturbance $\delta^{15}$N levels 6–10 years after fire. Here, we did not find similar short-term increases in $\delta^{15}$N after known prescribed fire events. It is uncertain whether higher resolution sampling using annual growth rings would provide greater insights into short-term effects of fire on wood $\delta^{15}$N, as N translocation may smooth out inter-annual disturbances to the N cycle.

It is very likely that fire-exclusion since 1964 has increased tree dominance, net N mineralization, and thus overall N availability in the no-fire stand (Reich et al., 2001), yet three out of four trees in this stand demonstrated significant declines in wood $\delta^{15}$N. In contrast, wood $\delta^{15}$N has remained stable throughout the 20th century in trees from the high-fire stand, despite prescribed fire that likely reduced N supply by decreasing rates of soil net nitrification and promoting further establishment of $C_4$ grasses in this stand (Reich et al., 2001). Stable wood $\delta^{15}$N in the high-fire stand could reflect a decline in N demand that was approximately proportional to declines in N supply, or that decreases in N supply due to fire were too small relative to stable N demand to cause a decline in $\delta^{15}$N values. Future studies should consider that wood $\delta^{15}$N patterns may result from changes in N supply driven directly by fire, in conjunction with changes in N demand driven indirectly by fire vis-à-vis

altered vegetation composition and abundance. Our results at this temperate oak savanna suggest an important influence of decadal-scale vegetation changes caused by fire on N cycling.

Recent studies have shown mixed results regarding the reliability of wood $\delta^{15}$N as a proxy for terrestrial N availability. In some settings wood $\delta^{15}$N appears to reliably integrate spatiotemporal variation in soil N supply relative to plant demand (Elmore et al., 2016; Kranabetter and Meeds, 2017; Sabo et al., 2020), whereas other results suggest an inconsistent relationship between wood $\delta^{15}$N and N cycling (Tomlinson et al., 2016; Burnham et al., 2019). Our study sought to clarify the relationships between N availability, wood $\delta^{15}$N, and long-term disturbance by sampling across four treatment stands to control for the presence and frequency of prescribed burning. Contrasting trends in heartwood $\delta^{15}$N across our treatment stands suggest minor roles for the heartwood-sapwood transition and exogenous drivers of wood $\delta^{15}$N at this study site. Nonetheless, the apparent lack of wood $\delta^{15}$N response to repeated burning at our study site, particularly in the high fire frequency treatment, raises uncertainty about the ability for individual tree-level $\delta^{15}$N to record the effects of low intensity fires on N cycling. We recommend that future studies of N cycling using wood $\delta^{15}$N utilize diligent site selection criteria to control for as many confounding factors as possible to help clarify the processes controlling variation in wood $\delta^{15}$N values.

## 5   Conclusions

In summary, over fifty years of long-term ecological research has provided a uniquely detailed understanding of the N cycle at CCESR to compare with our results. In contrast to our hypothesis, wood $\delta^{15}$N values were significantly different among stands before the onset of prescribed burning and differences in fire frequency did not correspond to long-term $\delta^{15}$N trends or temporally consistent shifts in mean $\delta^{15}$N across four treatment stands. Combined with findings from previous ecosystem biogeochemistry studies at CCESR, we interpret declining wood $\delta^{15}$N in the no-, low-, and medium-fire stands over the past 115 years to reflect declining N availability to mature oaks at this savanna. No direct effects of fire were consistently supported by our results; instead wood $\delta^{15}$N values were perhaps determined by changes in vegetation composition and abundance over time, due in part to early to mid 20th century fire suppression. Specifically, the encroachment of shrubs and juvenile trees in three stands likely increased N demand and competition for N, decreasing N availability for mature oaks, and influencing wood $\delta^{15}$N; although we note that such an interpretation is more plausible than conclusive.

*Author contributions.*  K.K.M, D.G., S.E.H., and P.B.R. secured funding. M.L.T., D.G., S.E.H., P.B.R., and K.K.M. developed the study design. Field sampling and tree-ring dating was conducted by M.L.T. and D.G, and isotopic analyses were done by D.M.N. M.L.T. performed the statistical analyses. M.L.T, D.G., and S.E.H. prepared the figures. All authors contributed to interpretation of the data. M.L.T. wrote the manuscript with editorial contributions from all authors.

*Competing interests.*  The authors declare no competing interests.

*Acknowledgements.* We thank the editor Dr. Edzo Veldkamp and reviewers Drs Peter Hietz and Rodica Pena, whose critiques and suggestions improved the manuscript. This work was supported by grants from the US National Science Foundation Long-Term Ecological Research Program (LTER) including DEB-0620652 and DEB-1234162. This research was also supported by NSF-DEB-1655148 to K.K.M., NSF-DEB-1655144 to D.G., the NSF REU program to M.L.T., and the University of Minnesota Undergraduate Research Opportunities Program
5   to M.L.T. Fire frequency treatments have been maintained with support of the Cedar Creek LTER program. Further support was provided by the Cedar Creek Ecosystem Science Reserve and the University of Minnesota. Daniel Ackerman, Kate Carlson, Daniel Crawford, Mara McPartland, and Madison Sherwood assisted with collecting tree core samples. Ryan Mattke and the UMN Borchert Map Library provided access to historical aerial imagery. Jack Dougherty assisted with tree core specimen preparation, and Robin Paulman conducted the isotopic analysis at CASIF. We thank Kurt Kipfmueller for helpful discussion.

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
