# Peer review of "Century-scale wood nitrogen isotope trajectories from an oak savanna with variable fire frequencies"

_Biogeosciences, 2020_

## Referee Comment (RC1) · Peter Hietz (Referee) · 20 May 2020

The biogeochemical cycle of nitrogen is complex and influenced by human activities in various ways. Understanding and monitoring the slow impact of human interference on N accumulation and losses is important, but limited by the lack of long-term records. A number of studies have therefore measured N15 signals in tree rings to investigate changes that may be interpreted as N availability or losses. The isotopic effects of various processes in the N cycle are variable and not perfectly understood, but reason-ably well to draw some general conclusion from differences in N15 signals, or changes thereof, in soils or plants. That said, interpreting any N15 trends in wood poses ad-

ditional questions and is not as straightforward as with C13. First N concentrations in wood are very low so precise isotopic analysis is more challenging than for C isotopes which means a higher signal-to-noise ratioSecond, at least in sapwood N is somewhat mobile so N isotope in a growth ring do not necessarily reflect the year the wood was formed. From the little that is known it is reasonably assumed that only a small proportion is translocated or not across many years. Third, while N in heartwood is likely limited to cell walls, in sapwood some is in living cells (and presumably more mobile) so concentrations in more recently produced wood tends to be higher (as seen in Fig. 5). Very little is known about related changes in N15 signals of wood. Fourth, changes in N15 might also carry a source signal (N15 trend in deposition). Finally, the discrimination during uptake and transfer in different species can be variable. Trends in N15 in wood are thus open to many interpretations unless some or all of the confounding factors can be controlled and studies such as the one by Tumper et al. that do so at least help to clarify how useful trends in tree ring N15 are.

While the study by Tumper et al. found a clear effect in sapwood on N concentrations, this appears to be unrelated to changes in wood N15, which are all seen in heartwood, so the inter-annual mobility and sapwood/heartwood differences are unlikely to be an issue. Given that the trees are from a small area, changes in N deposition and the N15 signal thereof at least should not affect differences in N15 trends. Also, they used only one species, so at least we should not expect differences in N15 found to be a species effects. That said, the previous documentation of species-specific trajectories (McLauchlan& Craine, 2012 Biogeosciences, 9, 867-874) where N15 can go up in one species while going down in another pours cold water on the use of tree ring N15 as an indicator of changes in the N cycle or availability and questions many interpretation of N15 trends one might suggest. Tumper et al. found significant differences in N15 trends between treatments. Frustratingly, the patterns seen do not offer an easy interpretation and a relationship with fire appears elusive. In the low-fire regime, wood N15 changes approx. with the onset of burns. However, N15 changed at the same time in the no-burn regime, in the medium-fire regime the shift in N15 was much earlier than the burning

regime, and in the high-frequency fire there was no change in N15 (Fig. 6). Having to reject their initial hypothesis, they discuss alternative causes of the observed N15 trends, specifically changes in vegetation. Still, if vegetation change has something to do with the fire regime, we would expect an effect to be somehow related to the time or intensity of the burns. Data on the change in vegetation cover might help to support or refute this idea, but are not presented. I am not sure if the aerial photographs available or other records could be analyzed for this.

A rather serious limitation of the study is that there are no replicates of fire treatment. The four trees per treatment were not from different plots, spatially quite close and at a substantial distance from trees sampled for other treatments. Unfortunately, there is a spatial variability in N15 as trees from different location differ substantially in wood N15 before any treatment. If this also affects N15 trends we do not know, but in the end the data do not convincingly show that N15 trends are related with fire regimes at all.

The results are presented in a useful way for readers to understand and make up their minds. For me, the clearest message of the manuscript is pointing to the challenges and perhaps limitations of using tree ring N15 and this as well as technical limitations should be acknowledge in the discussion.

Other than that, I suggest a number of relatively small changes or additions that might help interpret the data.

Table 2: please add the treatment (burn regimes) to the individual trees (this is in Table 1) or mark the groups of 4 in some way so that it is easy to compare groups.

Why is p>0.05 considered significant for the MK test for trends but p<0.01 for the Student t-test for shifts in mean?

Page 9/ line 12 "Standardizing wood 15N for each tree by subtracting the tree mean 15N from each data point produced similar results: standardized 15N values were different among stands in 1902 (p = 0.039)." If you standardize by subtracting the

mean, the standardized mean will be the same for all groups. If the values from 1902 are not the same (as the mean of the time series), this simply says that there were different trends, but this is better shown in the trends rather than some standardized values, so simply drop this.

"Tree age did not have an effect on the mean or trend of 15N." This is reassuring, but how was this tested and where is this shown? If there is a trend in individual trees, how do you distinguish an age from an environment effect? Looking at the data, age effect is a very implausible explanation for the trends found, which might be stated.

9/22 "Declines in wood 15N were roughly synchronous in these stands, beginning primarily between 1940 and 1965" I would not call events that differ by 20 years or more to be synchronous, not even roughly.

Wood % N was negatively correlated with wood N15 across all trees. This is potentially problematic: if there is a trend in %N that is caused by (recent) sapwood having higher N, and N and N15 are correlated, you might get trends in N15 that are related to changes in %N from heartwood to sapwood and not driven by external changes. Ideally, heartwood (the last 20 yrs or so) should thus be excluded. Given that changes in N15 occurred much earlier than 20 yrs ago, this appears not to be a problem, but should perhaps be mentioned in the discussion. In any case I suggest to test if the N $\sim$ N15 relationship also holds true if you exclude sapwood.

"Higher resolution sampling using annual or sub-annual growth rings may provide greater insights into short-term effects of fire on wood 15N and N availability and is recommended for future study" – I do not see how this would improve the outcome and would rather invest more analyses in sampling more trees and sites. Perhaps even pool wood across several years to dampen noise caused by short-term inter-annual variation, unless this is of specific interest (mostly not).

In the discussion studies that looked at N mineralization and total N at specific (and different) time points in the experiment are mentioned. These are published data, but

it would be useful to present them together with (perhaps only recent) wood N15 in a table or figure to see if there is any relationship.

---

## Referee Comment (RC2) · Rodica Pena (Referee) · 16 Jun 2020

The manuscript by Trumper et al. reports on a fascinating study examining the 15N enrichment in the savanna oak wood rings under the influence of fire events of different frequency. The basis of this study is the assumption that 15N levels in the wood are determined by the 15N enrichments of the taken-up N. Thus, interrogating the wood 15N enrichments, the authors may catch a glimpse of the effect of fire frequency on the soil N cycling or directly N availability in the soil. They demonstrated once more that the complex contributions of various processes in the nitrogen cycle make impossible assigning the same pattern of 15N enrichments to different trees/plots. This has been

[Figure]

**BGD**

the case, even if the trees were from plots that look similar in overstory vegetation, topography, and soil properties (1st part of the hypothesis). Maybe it depends a bit more on the soil properties. The authors may want to show the soil characteristics in the four plots. The 2nd part of the hypothesis, based on the assumption that soil N availability decreases (on long timescale) after burning, predicts a decrease in the wood 15N proportionally to the fire frequency. My major concern is that the experimental design does not enable precise statistical testing of this hypothesis. This is because there are no replicates for the treatments (only one plot with four trees) and the plots, localised at different distances, bring a high heterogeneity in the analysis anyhow. I do not think that the way the authors try to present the testing of this hypothesis is the most adequate. I am surprised by the regression analysis between a continuous (delta15N) and a categorical (Year)variable. I suggest evaluating the effect of time but no differences among treatments, using the trees as replicates in a repeated measured ANOVA. We all know the high variability among measurements that make the data difficult to meet the normality assumptions of ANOVA, but you can give a try. There is also a possibility to use GLMM and consider the tree as a random factor. I see in the text different p-values or expressions referring to the comparison among treatments (i.e., stands). I do not understand how were those comparisons done. I miss a table/figure presenting the mean value or min, median, max of data in each plot at each time point. The authors acknowledged that the fire differentially affects N soil cycle on a short vs long timescale. I have understood the reason, but I have missed the algorithm of their selection of the two samples per decade.

Specific comments Page 2-line 18: The main factor is the reduced losses that result in a lower enrichment of the soil N pools, the mycorrhizal transfer is added to that. Fig 1 & Fig 2: Fig 1 is almost identical with the figure from van der Sleen et al. 2017 (you probably need the copyright). The predictions shown in Fig. 2 are based on your box additions to the Fig 1. Maybe it would be sufficient enough to include those boxes as an inset in Fig 2. Page 3-lines 6-7: I think the soil N availability matters in those patterns. Maybe it is worth it to discuss the low N availability that is the case at CCESR. Page

6-line 19: a mean value of ring width would be helpful. Page 6-line 20: Is this an issue here? The inter-ring 15N mobility is relatively restricted to the youngest rings. Page 8-line 20: the regime shift detection analysis is less known to some readers. Could you please add a short explanatory phrase. Page 9-line 3: In the case you used a nonparametric test, the median value is more appropriate to be shown. Page 9-lines 10-20: please specify where are all these data displayed. Figure 5: please also include the heartwood-sapwood transition in the left panel. Page 12-line 27: The correlation figure between N and delta15N could be of interest as the negative correlation is a bit unexpected. Page 13-line 1: but see Meerts 2002(doi.org/10.1051/forest:2002059) and Martin et al. 2014 (doi.org/10.1111/nph.12943).

---

## Author Comment (AC1) · 2 Jul 2020

We are grateful to the editor Dr. Edzo Veldkamp and reviewers Drs. Hietz and Pena for their time, expertise, and thoughtful input on our manuscript. We have worked diligently to carefully consider each of the many individual suggestions, and in nearly all cases we incorporated revisions accordingly toward what we believe is an improved manuscript. In the few cases where we disagreed with the reviewer critiques, we modified the manuscript to articulate our rationale and interpretation with greater clarity and expanded justification, and specific rebuttals are detailed in the responses to reviewers text below. Based on these responses to reviewers, and the manuscript changes we propose, as articulated below, we hope the editor will find the revised manuscript to be suitable for publication.

Below, the reviewer comments are copied verbatim in black. We have responded to each comment below in blue, and summarized specific changes made to the manuscript in green.

**Author Responses to Peter Hietz (Referee)**
**(Responses in Blue) (Summary of revisions in green)**

The biogeochemical cycle of nitrogen is complex and influenced by human activities in various ways. Understanding and monitoring the slow impact of human interference on N accumulation and losses is important, but limited by the lack of long-term records. A number of studies have therefore measured N15 signals in tree rings to investigate changes that may be interpreted as N availability or losses. The isotopic effects of various processes in the N cycle are variable and not perfectly understood, but reasonably well to draw some general conclusion from differences in N15 signals, or changes thereof, in soils or plants. That said, interpreting any N15 trends in wood poses additional questions and is not as straightforward as with C13. First N concentrations in wood are very low so precise isotopic analysis is more challenging than for C isotopes which means a higher signal-to-noise ratio. Second, at least in sapwood N is somewhat mobile so N isotope in a growth ring do not necessarily reflect the year the wood was formed. From the little that is known it is reasonably assumed that only a small proportion is translocated or not across many years. Third, while N in heartwood is likely limited to cell walls, in sapwood some is in living cells (and presumably more mobile) so concentrations in more recently produced wood tends to be higher (as seen in Fig. 5). Very little is known about related changes in N15 signals of wood. Fourth, changes in N15 might also carry a source signal (N15 trend in deposition). Finally, the discrimination during uptake and transfer in different species can be variable. Trends in N15 in wood are thus open to many interpretations unless some or all of the confounding factors can be controlled and studies such as the one by Tumper et al. that do so at least help to clarify how useful trends in tree ring N15 are.

We are grateful for the careful and encouraging review provided by Dr. Hietz.

While the study by Tumper et al. found a clear effect in sapwood on N concentrations, this appears to be unrelated to changes in wood N15, which are all seen in heartwood, so the inter-annual mobility and sapwood/heartwood differences are unlikely to be an issue. Given that the trees are from a small area, changes in N deposition and the N15 signal thereof at least should not affect differences in N15 trends. Also, they used only one species, so at least we should not expect differences in N15 found to be a species effects. That said, the previous documentation of species-specific trajectories (McLauchlan & Craine, 2012 Biogeosciences, 9, 867-874) where N15 can go up in one species while going down in another pours cold water on the use of tree ring N15 as an indicator of changes in the N cycle or availability and questions many interpretation of N15 trends one might suggest. Tumper et al. found significant differences in N15 trends between treatments. Frustratingly, the patterns seen do not offer an easy interpretation and a relationship with fire appears elusive. In the low-fire regime, wood N15 changes approx. with the onset of burns. However, N15 changed at the same time in the no-burn regime, in the medium-fire regime the shift in N15 was much earlier than the burning regime, and in the high-frequency fire there was no change in N15 (Fig. 6). Having to reject their initial hypothesis, they discuss alternative causes of the observed N15 trends, specifically changes in vegetation. Still, if vegetation change has something to do with the fire regime, we would expect an effect to be somehow related to the time or intensity of the burns. Data on the change in vegetation cover might help to support or refute this idea, but are not presented. I am not sure if the aerial photographs available or other records could be analyzed for this.

Historical aerial photos are available for our study area for the years 1938, 1954, 1973, 1991, 2015, and 2016.  While we did not conduct a quantitative analysis of landcover as evidenced in these photos, we did conduct a thorough qualitative interpretation of them. For simplicity, we elected to only include the end member years, 1938 and 2016, in Figure 3. Images from the intervening years, not shown, illustrate the intermediate steps of the patterns we discuss in the text, whereby canopy density increased through time in the no-fire, low-fire, and medium-fire stands but remained relatively constant in the high-fire stand. Our results are in general agreement with the results from Peterson and Reich (2001) at our site and Faber-Langendoen and Davis (1995) from an adjacent experimental site. We have added this reference and expanded our interpretation in the manuscript accordingly:

We revised paragraph two of the methods to read: "Although systematic vegetation inventories were not conducted prior to the onset of prescribed burning, a series of historical aerial photographs are available for our study area between the years 1938 and 2016 (McAuliffe et al., 2017; MHAPO, 2019). Our qualitative interpretation of these photographs indicates that the savanna vegetation prior to fire treatment at CCESR consisted of mixed woodland and grass communities (Fig. 2). Over time, oak tree vegetation – predominantly bur oak, *Quercus macrocarpa*, and northern pin oak, *Quercus ellipsoidalis* – increasingly dominated fire-excluded stands, while $C_4$ grasses and sedges increased in abundance in frequently burned stands (Peterson and Reich, 2001; Dijkstra et al., 2006)."

We revised paragraph five of the discussion to read: "Our qualitative assessment of aerial photographs between 1938 and 2016 indicates that canopy density increased through time in the no-, low-, and medium-fire stands but remained relatively constant in the high-fire stand. Our interpretation is in general agreement with results from other studies at CCESR (Peterson and Reich, 2001) and an adjacent experimental site (Faber-Langendoen and Davis, 1995)."

A rather serious limitation of the study is that there are no replicates of fire treatment. The four trees per treatment were not from different plots, spatially quite close and at a substantial distance from trees sampled for other treatments. Unfortunately, there is a spatial variability in N15 as trees from different location differ substantially in wood N15 before any treatment. If this also affects N15 trends we do not know, but in the end the data do not convincingly show that N15 trends are related with fire regimes at all.

We agree that the lack of fire treatment replicates imposes a limitation on the interpretation. With the limited resources available for this research project, we made a conscious decision in designing the study to maximize the temporal extent of sampling and evaluate $\delta^{15}$N across a range of burn frequency treatments. We did not know beforehand how much variation in $\delta^{15}$N values would exist among trees with treatment units. We still believe our sampling design represents a valuable contribution to the literature because it is among the most spatially and temporally dense sampling of tree-ring $\delta^{15}$N at a single site.

The rationale of our study design was not adequately articulated in the manuscript. Knowing what we do now, we would advocate that a future study should (1) include replicates of fire treatment and (2) sample trees that were more spread out within treatments. We have expanded the manuscript text with discussion of this limitation:

We added the following text in paragraph six of the discussion: "The relative close proximity of sampled trees within treatment stands may also contribute to the ambiguous relationship between fire and wood $\delta^{15}$N at this site. The four trees sampled per burn stand were spatially quite close to each other (on average 34 meters between trees within each burn unit) and we were unable to replicate sampling in other units with the same fire treatment. Given the spatial variability in wood $\delta^{15}$N that existed before the fire treatment began, additional replicates of fire treatment may help to clarify the relationship between fire and wood $\delta^{15}$N."

The results are presented in a useful way for readers to understand and make up their minds. For me, the clearest message of the manuscript is pointing to the challenges and perhaps limitations of using tree ring N15 and this as well as technical limitations should be acknowledge in the discussion.

We agree that the challenges and limitations of using wood $\delta^{15}$N were not made explicit in our original submission. To clarify for the reader and future related studies, we now acknowledge these issues in the discussion section:

We added the paragraph to the discussion: "Recent studies have shown mixed results regarding the reliability of wood $\delta^{15}$N as a proxy for terrestrial N availability. In some settings wood $\delta^{15}$N appears to reliably integrate spatiotemporal variation in soil N supply relative to plant demand (Elmore et al., 2016; Kranabetter and Meeds, 2017; Sabo et al., 2020), whereas other results suggest an inconsistent relationship between wood $\delta^{15}$N and N cycling (Tomlinson et al., 2016; Burnham et al., 2019). Our study sought to clarify the relationships between N availability, wood $\delta^{15}$N, and long-term disturbance by sampling across four treatment stands to control for the presence and frequency of prescribed burning. Contrasting trends in heartwood $\delta^{15}$N across our treatment stands suggest minor roles for the heartwood-sapwood transition and exogenous drivers of wood $\delta^{15}$N at this study site. Nonetheless, the apparent lack of wood $\delta^{15}$N response to repeated burning at our study site, particularly in the high fire frequency treatment, raises uncertainty about the ability for individual tree-level $\delta^{15}$N to record the effects of low intensity fires on N cycling. We recommend that future studies of N cycling using wood $\delta^{15}$N utilize diligent site selection criteria to control for as many confounding factors as possible to help clarify the processes controlling variation in wood $\delta^{15}$N values."

Other than that, I suggest a number of relatively small changes or additions that might help interpret the data.

Table 2: please add the treatment (burn regimes) to the individual trees (this is in Table 1) or mark the groups of 4 in some way so that it is easy to compare groups.

Table 2 has been revised to include this information.

Why is p>0.05 considered significant for the MK test for trends but p<0.01 for the Student t-test for shifts in mean?

Thank you for highlighting this unintended inconsistency. We have revised the significance level for the MK test to p<0.01. Testing for significant increases in wood %N at p<0.01 yields similar results: wood %N increased significantly in 11 out of 16 trees at p<0.01 compared to 12 out of 16 trees at p<0.05.

Page 9/ line 12 "Standardizing wood 15N for each tree by subtracting the tree mean 15N from each data point produced similar results: standardized 15N values were different among stands in 1902 (p = 0.039)." If you standardize by subtracting the mean, the standardized mean will be the same for all groups. If the values from 1902 are not the same (as the mean of the time series), this simply says that there were different trends, but this is better shown in the trends rather than some standardized values, so simply drop this.

This sentence was a carryover from an early version of the manuscript, attempting to clarify methods relative to a few other key studies. We feel this point is superfluous and we have dropped this sentence from the manuscript.

"Tree age did not have an effect on the mean or trend of 15N." This is reassuring, but how was this tested and where is this shown? If there is a trend in individual trees, how do you distinguish an age from an environment effect? Looking at the data, age effect is a very implausible explanation for the trends found, which might be stated.

Thank you for raising this point, which was not sufficiently outlined in the manuscript. As the reviewer implies and the data indicate, the age effect is an implausible explanation for the wood $\delta^{15}$N trajectories. The temporal wood $\delta^{15}$N patterns in Fig. 6a-c do not appear as a gradual age-related effect, but rather as relatively invariant values, a shift to lower values, followed again by relatively invariant values. Additionally, Fig. 6d does not show any trends that could be related to tree age.

We have added the following in paragraph one of the results section: "Two approaches revealed that tree age likely did not have an effect on the mean or trend of $\delta^{15}$N. First, we compared wood $\delta^{15}$N with tree biological age rather than Gregorian calendar year using inner ring dates to estimate tree age. Tree age and $\delta^{15}$N were negatively correlated (r = -0.35; $p$ < 0.001), however this negative relationship may be expected given the negative temporal trends in $\delta^{15}$N in the majority of trees. Additionally, we tested for an age effect on wood $\delta^{15}$N by dividing trees into four age classes based on ring counts (Table 1, Table 2). A Kruskal-Wallis test of wood $\delta^{15}$N across age classes revealed that tree age did not have a significant effect on the mean ranks of $\delta^{15}$N ($p$ = 0.242)."

Table 1 and Table 2 were revised to include tree age class information with the following description: "Trees in age class 1 contain <129 rings, trees in age class 2 contain 130-159 rings, trees in age class 3 contain 160-189 rings, and trees in age class 4 contain > 190 rings".

9/22 "Declines in wood 15N were roughly synchronous in these stands, beginning primarily between 1940 and 1965" I would not call events that differ by 20 years or more to be synchronous, not even roughly.

We agree. We followed the suggestion of the reviewer by removing mention of synchrony here and throughout the manuscript.

Wood % N was negatively correlated with wood N15 across all trees. This is potentially problematic: if there is a trend in %N that is caused by (recent) sapwood having higher N, and N and N15 are correlated, you might get trends in N15 that are related to changes in %N from heartwood to sapwood and not driven by external changes. Ideally, heartwood (the last 20 yrs or so) should thus be excluded. Given that changes in N15 occurred much earlier than 20 yrs ago, this appears not to be a problem, but should perhaps be mentioned in the discussion. In any case I suggest to test if the N ~ N15 relationship also holds true if you exclude sapwood.

We further examined the relationship between wood $\delta^{15}$N and %N, as suggested.

We added the sentence in paragraph seven of the results: "Excluding all samples from the sapwood, wood % N and wood $\delta^{15}$N remained negatively correlated, statistically significant although the relationship weakened (r = -0.30; p < 0.001)."

We added the sentences to paragraph two of the discussion: "Wood % N and $\delta^{15}$N were negatively correlated across all samples and this negative relationship diminished when excluding samples from the sapwood. Although we do not have further evidence to suggest that this is a meaningful relationship, future studies should examine relationships between wood % N and $\delta^{15}$N because their covariance could be problematic if trends in $\delta^{15}$N are strongly related to % N in addition to external factors."

"Higher resolution sampling using annual or sub-annual growth rings may provide greater insights into short-term effects of fire on wood 15N and N availability and is recommended for future study" – I do not see how this would improve the outcome and would rather invest more analyses in sampling more trees and sites. Perhaps even pool wood across several years to dampen noise caused by short-term inter-annual variation, unless this is of specific interest (mostly not).

We agree with this critique. Higher resolution sampling may provide greater insights into the short-term effects of fire, although there is substantial uncertainty surrounding this question of N translocation. Translocation of N likely limits the extent to which annual wood $\delta^{15}$N patterns reflect inter-annual disturbances to the N cycle. We clarify in the manuscript:

We added the following sentences in paragraph seven of the discussion: "It is uncertain whether higher resolution sampling using annual growth rings would provide greater insights into short-term effects of fire on wood $\delta^{15}$N, as N translocation may smooth out inter-annual disturbances to the N cycle."

In the discussion studies that looked at N mineralization and total N at specific (and different) time points in the experiment are mentioned. These are published data, but it would be useful to present them together with (perhaps only recent) wood N15 in a table or figure to see if there is any relationship.

Thank you for raising this point, we agree that this is a data rich site and it would be beneficial to add more data for comparison. We included a new table with data from Reich et al. (2001) who examined net N mineralization rates in three of four stands that we sampled in 1995, as well as data from Pellegrini et al. (2020) who gathered soil % N and soil $\delta^{15}$N data at multiple soil depths in two of four stands that we sampled in 2016. Although these additional data add context to our results, they ultimately do not clarify interpretation of our observed wood $\delta^{15}$N patterns.

| Year | Data | Soil Depth (cm) | Low-fire | Medium-fire | High-fire |
|---|---|---|---|---|---|
| 1992 | Wood $\delta^{15}$N | NA | -2.46 | -1.06 | 0.13 |
| 1995 | Net N mineralization | 0–15 | 7.21 | 10.42 | 2.31 |
| 1997 | Wood $\delta^{15}$N | NA | -1.84 | -1.94 | -0.76 |
| 2016 | % N | 0–5 | 0.12 | NA | 0.1 |
| 2016 | % N | 5–10 | 0.06 | NA | 0.06 |
| 2016 | % N | 10–20 | 0.05 | NA | 0.06 |
| 2016 | % N | 20–60 | 0.02 | NA | 0.04 |
| 2016 | % N | 60–100 | 0.01 | NA | 0.03 |
| 2016 | $\delta^{15}$N | 0–5 | 1.13 | NA | 2.25 |
| 2016 | $\delta^{15}$N | 5–10 | 4.15 | NA | 3.57 |
| 2016 | $\delta^{15}$N | 10–20 | 4.76 | NA | 3.85 |
| 2016 | $\delta^{15}$N | 20–60 | NA | NA | 5.14 |
| 2016 | $\delta^{15}$N | 60–100 | NA | NA | 2.93 |
| 2017 | Wood $\delta^{15}$N | NA | -2.54 | -1.58 | -0.32 |

Table 4. Comparison of previously collected soil N data and our wood $\delta^{15}$N data from CCESR. Values represent the mean of all samples within each burn unit. Net N mineralization data from Reich et al. (2001), and soil % N and $\delta^{15}$N data from Pellegrini et al. (2020).

We revised paragraph four in the discussion to reflect these additional data. This paragraph now reads: "Prior studies at CCESR have examined N cycling in the same stands we sampled for wood $\delta^{15}$N across the burn experiment. Reich et al. (2001) measured net N mineralization in 1995 in three of the four stands we sampled (low-, medium-, and high-fire). Net N mineralization decreased nonlinearly with increasing fire frequency among these stands. However, our wood $\delta^{15}$N samples in 1992 and 1997 increased with increasing fire frequency among these stands. Wood $\delta^{15}$N values from 1992 and 1997 were lower in the low- and medium-fire stands compared to the high-fire stand, despite relatively higher net N mineralization in the former stands in 1995 (Table 4; Reich et al. 2001). This could point to large inter-annual variation in wood $\delta^{15}$N

or that soil net N mineralization may not be strongly related to wood $\delta^{15}$N at our study site. More recently, Pellegrini et al. (2020) measured soil $\delta^{15}$N and cumulative N stocks at multiple depths in two of the four stands we sampled (low- and high-fire) in 2016. They found $\delta^{15}$N decreased with more frequent burning and $\delta^{15}$N increased with depth across the entire fire frequency gradient, while fire did not have a clear effect on total soil N between the low- and high-fire frequency stands we sampled (Pellegrini et al., 2020). In contrast, our wood $\delta^{15}$N in 2017 was significantly lower in the low-fire stand compared to the high-fire stand and we found limited evidence for variation in wood $\delta^{15}$N related to long-term fire frequency. Altogether, these nuanced results from previous studies add context but do not clarify interpretation of our observed wood $\delta^{15}$N patterns."

**Author Responses to Rodica Pena (Referee)**
**(Responses in Blue) (Summary of revisions in green)**

The manuscript by Trumper et al. reports on a fascinating study examining the 15N enrichment in the savanna oak wood rings under the influence of fire events of different frequency. The basis of this study is the assumption that 15N levels in the wood are determined by the 15N enrichments of the taken-up N. Thus, interrogating the wood 15N enrichments, the authors may catch a glimpse of the effect of fire frequency on the soil N cycling or directly N availability in the soil. They demonstrated once more that the complex contributions of various processes in the nitrogen cycle make impossible assigning the same pattern of 15N enrichments to different trees/plots. This has been the case, even if the trees were from plots that look similar in overstory vegetation, topography, and soil properties (1st part of the hypothesis). Maybe it depends a bit more on the soil properties. The authors may want to show the soil characteristics in the four plots.

We are grateful for the careful and encouraging review provided by Dr. Pena.

We have added some clarifying description of the soil characteristics in the four plots:

In the revisions, we added the following sentences in paragraph three of the methods: "Trees in the low-fire stand grew on Sartell fine sand mapped in soil surveys with 0–6% slopes and trees in no-, medium-, and high-fire stands grew on Sartell fine sand mapped in soil surveys with 6–15% slopes. In the localized vicinity of the trees sampled across these sites, slopes rarely exceeded 8%. The Sartell series consists of excessively drained soils that are rapidly permeable and have low available N, low organic matter content, and low available water capacity (Grigal et al., 1974). The upper 15 cm of Sartell fine sand has a pH of 5.3, 0.025% total nitrogen (18 µmol/g dry soil), and ~0.3% organic matter (Grigal et al., 1974; Tilman, 1984)."

The 2nd part of the hypothesis, based on the assumption that soil N availability decreases (on long timescale) after burning, predicts a decrease in the wood 15N proportionally to the fire frequency. My major concern is that the experimental design does not enable precise statistical testing of this hypothesis. This is because there are

no replicates for the treatments (only one plot with four trees) and the plots, localised at different distances, bring a high heterogeneity in the analysis anyhow. I do not think that the way the authors try to present the testing of this hypothesis is the most adequate. I am surprised by the regression analysis between a continuous (delta15N) and a categorical (Year)variable. I suggest evaluating the effect of time but no differences among treatments, using the trees as replicates in a repeated measured ANOVA. We all know the high variability among measurements that make the data difficult to meet the normality assumptions of ANOVA, but you can give a try. There is also a possibility to use GLMM and consider the tree as a random factor.I see in the text different p-values or expressions referring to the comparison among treatments (i.e., stands). I do not understand how were those comparisons done.

We agree that the lack of fire treatment replicates is a limitation. We addressed this concern, as described above in response to a similar comment from the first reviewer. In terms of regression with a categorical variable, this is not uncommon in the literature (Cottingham et al., 2005; McLauchlan and Craine, 2012; Howard and McLauchlan, 2015). Furthermore, we believe that our nonparametric analytical approach is both conservative and pragmatic in terms of expectations for evaluation of results. We describe our approach to post-hoc testing in greater detail in the revised version:

We added the following sentences to paragraph one of the results: "A Kruskal-Wallis test of wood $\delta^{15}$N across the time series for the four stands revealed that $\delta^{15}$N was significantly different among stands ($p < 0.001$). We used the nonparametric Games-Howell test for post-hoc analysis due to unequal variances and group sizes between burn treatment stands. The Games-Howell test revealed that $\delta^{15}$N in the no-fire stand was lower than all other stands ($p < 0.001$)."

I miss a table/figure presenting the mean value or min, median, max of data in each plot at each time point.

While we strived to ensure that the time series plots represented clearly legible values for individual and population data (including minimum and maximum values), we have created a new table of mean wood $\delta^{15}$N in each plot at each time point.

| Year | No-fire mean $\delta^{15}$N | Low-fire mean $\delta^{15}$N | Medium-fire mean $\delta^{15}$N | High-fire mean $\delta^{15}$N |
|---|---|---|---|---|
| 1902 | -2.19* | 0.38 | 1.65 | -1.16 |
| 1907 | -2.91* | 0.54 | 2.21 | -1.58 |
| 1912 | -1.14* | 0.77 | 1.93 | -0.98 |
| 1917 | -0.03 | 1.19 | 1.94 | -0.46 |

| | | | |
|------|------|------|------|
| 1922 | -1.81 | 0.86 | 1.32 | -1.24 |
| 1927 | -0.7 | 0.98 | 1.37 | -0.09 |
| 1932 | -1.24 | 1.17 | 1.37 | -0.53 |
| 1937 | -0.65 | 0.94 | 1.25 | -0.66 |
| 1942 | -1.06 | 0.07 | 0.9 | -0.62 |
| 1947 | -1.27 | 1.26 | 0.26 | -0.5 |
| 1952 | -1.54 | -0.18 | -0.68 | -0.72 |
| 1957 | -1.63 | 0.05 | -0.35 | -0.09 |
| 1962 | -0.9 | -0.4 | -0.94 | -0.71 |
| 1967 | -2.34 | -2.2 | -2.07 | -0.6 |
| 1972 | -3.12 | -2.35 | -2.06 | -0.47 |
| 1977 | -3.15 | -2.54 | -1.94 | -0.67 |
| 1982 | -2.81 | -2.09 | -2.14 | -0.91 |
| 1987 | -2.6 | -2.45 | -2.25 | -1.24 |
| 1992 | -3.08 | -3.38 | -1.9 | -1.32 |
| 1997 | -3.28 | -2.65 | -2.29 | -1.33 |
| 2002 | -3.3 | -3.25 | -2.44 | -1.15 |
| 2007 | -3.2 | -3.04 | -2.41 | -0.93 |
| 2012 | -3.27 | -3.16 | -2.54 | -0.83 |
| 2017 | -3.53 | -2.87 | -2.16 | -1.03 |

Table 3. Mean wood $\delta^{15}N$ (‰) by year in each burn unit. Mean values * indicate years with only one $\delta^{15}N$ sample in that year.

The authors acknowledged that the fire differentially affects N soil cycle on a short vs long timescale. I have understood the reason, but I have missed the algorithm of their selection of the two samples per decade.

In order to clarify our rationale, we have added text, as indicated below. Our decision to select two samples per decade was motivated by two factors. First, we had to optimize the spatial and temporal coverage of sampling against analytical cost constraints. Choosing two wood $\delta^{15}$N samples per decade rather than annual sampling allowed us to expand our temporal coverage and sample more trees. Second, N translocation likely inhibits true annual resolution of wood $\delta^{15}$N as a proxy of N availability. Two samples per decade therefore seemed like a reasonable trade-off between temporal resolution and the length and spatial coverage of the record.

This section in paragraph four of the methods was revised to read: "Although wood samples were partitioned at annual resolution, we did not analyze all wood samples for $\delta^{15}$N due to cost and time constraints. Rather, we selected two wood samples per decade for $\delta^{15}$N measurement. The decision to forego annual resolution and analyze our data at supra-annual timescales also aimed to mitigate the known phenomenon of inter-ring mobility of N-containing compounds that could smooth out inter-annual variation in $\delta^{15}$N (Hart and Classen, 2003; McLauchlan et al., 2017)."

Specific comments Page 2-line 18: The main factor is the reduced losses that result in a lower enrichment of the soil N pools, the mycorrhizal transfer is added to that.

We agree with this point and the primary importance of reduced N losses was not sufficiently expressed in this section. We propose the following revisions:

This section in paragraph two of the introduction now reads: "When N availability is low, reduced N losses result in lower $\delta^{15}$N values of remaining N pools. In addition, plants are more likely to receive N from mycorrhizal fungi than from direct uptake from inorganic N pools; mycorrhizal fungi are known to provide N with relatively low $\delta^{15}$N values to plants (Hobbie and Högberg, 2012)."

Fig 1 & Fig 2: Fig 1 is almost identical with the figure from van der Sleen et al. 2017 (you probably need the copyright).The predictions shown in Fig. 2 are based on your box additions to the Fig 1. Maybe it would be sufficient enough to include those boxes as an inset in Fig 2.

We agree that Fig. 1 of the manuscript is quite similar to the figure from van der Sleen et al. (2017). Rather than pursue copyright permission, we removed Fig. 1 from the manuscript. In addition, we moved the boxes from the old Fig. 1 to the new Fig. 1. These steps address both suggestions from the reviewer and we believe that the relevant fire-$\delta^{15}$N processes are summarized well in the new Fig. 1.

The revised Fig. 1 is shown below:

[Figure]

**Figure 1:** Hypothesized effects of fire experimentation on wood $\delta^{15}$N values relative to the historical fire frequency (FF). Increased FF would lead to higher relative wood $\delta^{15}$N values if combustion and isotopic fractionation with N volatilization dominate (A). In contrast, increased FF would lead to lower relative wood $\delta^{15}$N values if N cycling effects such as reduced N stocks and non-fire N losses dominate the trajectory of wood $\delta^{15}$N values (B). Colored boxes indicate potential contrasting effects of fire frequency on wood $\delta^{15}$N values. Pink and blue boxes indicate processes resulting from higher and lower FF, respectively.

Page 3-lines 6-7: I think the soil N availability matters in those patterns. Maybe it is worth it to discuss the low N availability that is the case at CCESR.

Additional information regarding the soil N status at CCESR was included in the same revised section (above) that describes soil characteristics in more detail.

The following sentences were added in paragraph three of the methods: "The Sartell series consists of excessively drained soils that are rapidly permeable and have low available N, low organic matter content, and low available water capacity (Grigal et al., 1974). The upper 15 cm of Sartell fine sand has a pH of 5.3, 0.025% total nitrogen (18 µmol/g dry soil), and ~0.3% organic matter (Grigal et al., 1974; Tilman, 1984)."

Page 6-line 19: a mean value of ring width would be helpful.

This was added.

The following sentence was added in paragraph four of the methods: "Mean ring width from 1902–2017 across all N cores was 1.11 mm."

Page 6-line 20: Is this an issue here? The inter-ring 15N mobility is relatively restricted to the youngest rings.

Although inter-ring $\delta^{15}$N mobility is relatively restricted to the youngest rings, we believe that this remains an issue because all of the rings were once young and therefore influenced by $\delta^{15}$N mobility.

Page 8-line 20: the regime shift detection analysis is less known to some readers. Could you please add a short explanatory phrase.

Thank you for bringing this to our attention. We added a short clarification to the manuscript.

We added the sentence in paragraph six of the methods: "This method applies sequential $t$-tests to time series data to statistically identify shifts in the mean state (Rodionov, 2004)."

Page 9-line 3: In the case you used a nonparametric test, the median value is more appropriate to be shown.

The regime shift algorithm we used was a parametric test, therefore the mean value is appropriate. Although we chose nonparametric tests for testing significant mean differences and trend, we chose a parametric test for regime shift detection because this method has been shown to be quite robust to violations of the normality assumption of the data (Rodionov, 2004). We also felt that the apparent mean states in our data suggested that a test for mean differences was more appropriate than a piecewise regression approach.

Page 9-lines 10-20: please specify where are all these data displayed.

These data were not displayed in our original submission. Rather than creating a new figure, we more explicitly outline our post-hoc tests for $\delta^{15}$N differences between burn treatment stands. We also summarize the corresponding range of p-values.

We added the following sentences to paragraph one of the results: "We used the nonparametric Games-Howell test for post-hoc analysis due to unequal variances and group sizes between burn treatment stands. The Games-Howell test revealed that $\delta^{15}$N in the no-fire stand was lower than all other stands ($p < 0.001$). In contrast, the Games-Howell test revealed nonsignificant differences in $\delta^{15}$N between the low- and medium-fire stands ($p = 0.385$), low- and high-stands ($p = 0.944$), and medium- and high-fire stands ($p = 0.441$)."

Figure 5: please also include the heartwood-sapwood transition in the left panel.

We revised this figure accordingly.

Page 12-line 27: The correlation figure between N and delta15N could be of interest as the negative correlation is a bit unexpected.

This was addressed above in response to a similar comment from the other reviewer.

Page 13-line 1: but see Meerts 2002(doi.org/10.1051/forest:2002059) and Martin et al. 2014 (doi.org/10.1111/nph.12943).

We appreciate the reviewer for bringing these papers to our attention. Our manuscript shows that wood % N increased in the sapwood in all individual trees. We suggested that this result matches well documented patterns in the literature. However, these papers shared by the reviewer show insignificant differences between heartwood % N and sapwood % N, indicating more mixed results in the literature. We revised the manuscript to reflect the uncertainty regarding wood % N variation within trees.

We revised paragraph two of the discussion to read: "In contrast to wood $\delta^{15}$N, wood % N trajectories were positive and consistent among stands. Wood % N is not as commonly reported in wood N isotope studies because this metric likely incorporates tree physiology and thus may not reflect ecosystem N cycling (Gerhart and McLauchlan, 2014). Positive wood % N trajectories shown here match documented patterns of increasing wood % N after the heartwood-sapwood transition in the wood $\delta^{15}$N literature (Gerhart and McLauchlan, 2014), yet this pattern has seen more mixed results in the wood chemistry literature (Meerts, 2002; Martin et al., 2014)."

References

Burnham, M. B., Adams, M. B., & Peterjohn, W. T. (2019). Assessing tree ring δ15N of four temperate deciduous species as an indicator of N availability using independent long-term records at the Fernow Experimental Forest, WV. *Oecologia*, *191*(4), 971–981. https://doi.org/10.1007/s00442-019-04528-4

Cottingham, K. L., Lennon, J. T., & Brown, B. L. (2005). Knowing when to draw the line: designing more informative ecological experiments. *Frontiers in Ecology and the Environment*, *3*(3), 145–152. https://doi.org/10.1890/1540-9295(2005)003[0145:KWTDTL]2.0.CO;2

Elmore, A. J., Nelson, D. M., & Craine, J. M. (2016). Earlier springs are causing reduced nitrogen availability in North American eastern deciduous forests. *Nature Plants*, *2*(10), 1–5. https://doi.org/10.1038/nplants.2016.133

Faber-Langendoen, D., & Davis, M. A. (1995). Effects of Fire Frequency on Tree Canopy Cover at Allison-Savanna, Eastcentral Minnesota, USA. *Natural Areas Journal*.

Kranabetter, J. M., & Meeds, J. A. (2017). Tree ring δ15N as validation of space-for-time substitution in disturbance studies of forest nitrogen status. *Biogeochemistry*, *134*(1–2), 201–215. https://doi.org/10.1007/s10533-017-0355-4

Martin, A. R., Erickson, D. L., Kress, W. J., & Thomas, S. C. (2014). Wood nitrogen concentrations in tropical trees: phylogenetic patterns and ecological correlates. *New Phytologist*, *204*(3), 484–495. https://doi.org/10.1111/nph.12943

McAuliffe, C. P., Lage, K., & Mattke, R. (2017). Access to Online Historical Aerial Photography Collections: Past Practice, Present State, and Future Opportunities. *Journal of Map and Geography Libraries*, *13*(2), 198–221. https://doi.org/10.1080/15420353.2017.1334252

Meerts, P. (2002). Mineral nutrient concentrations in sapwood and heartwood: a literature review. *Annals of Forest Science*, *59*(7), 713–722. https://doi.org/10.1051/forest:2002059

Pellegrini, A. F. A., McLauchlan, K. K., Hobbie, S. E., Mack, M. C., Marcotte, A. L., Nelson, D. M., ... Whittinghill, K. (2020). Data from: Frequent burning causes large losses of carbon from deep soil layers in a tem- perate savanna. *Dryad Digital Repository*. https://doi.org/10.5061/dryad.02v6wwq07

Sabo, R. D., Elmore, A. J., Nelson, D. M., Clark, C. M., Fisher, T., & Eshleman, K. N. (2020). Positive correlation between wood $\delta^{15}N$ and stream nitrate concentrations in two temperate deciduous forests. *Environmental Research Communications*, *2*(2), 025003. https://doi.org/10.1088/2515-7620/AB77F8

Tomlinson, G., Buchmann, N., Siegwolf, R., Weber, P., Thimonier, A., Pannatier, E. G., … Waldner, P. (2016). Can tree-ring δ15N be used as a proxy for foliar δ15N in European beech and Norway spruce? *Trees - Structure and Function*, *30*(3), 627–638. https://doi.org/10.1007/s00468-015-1305-1